# Activation-Guided Consensus Merging for Large Language Models

**Yuxuan Yao**[1,2], **Shuqi Liu**[3], **Zehua Liu**[3], **Qintong Li**[4], **Mingyang Liu**[1,2], **Xiongwei Han**[3],
**Zhijiang Guo**[5,6], **Han Wu**[3†], **Linqi Song**[1,2†]

[1]Department of Computer Science, City University of Hong Kong
[2]City University of Hong Kong Shenzhen Research Institute
[3]Huawei Noah's Ark Lab, Hong Kong SAR
[4]University of Hong Kong
[5]Hong Kong University of Science and Technology (Guangzhou)
[6]Hong Kong University of Science and Technology
`(yuxuanyao3-c@my., linqi.song@)cityu.edu.hk`
`wu.han1@huawei.com`

## Abstract

Recent research has increasingly focused on reconciling the reasoning capabilities of System 2 with the efficiency of System 1. While existing training-based and prompt-based approaches face significant challenges in terms of efficiency and stability, model merging emerges as a promising strategy to integrate the diverse capabilities of different Large Language Models (LLMs) into a unified model. However, conventional model merging methods often assume uniform importance across layers, overlooking the functional heterogeneity inherent in neural components. To address this limitation, we propose **A**ctivation-Guided **C**onsensus **M**erging (**ACM**), a plug-and-play merging framework that determines layer-specific merging coefficients based on mutual information between activations of pre-trained and fine-tuned models. ACM effectively preserves task-specific capabilities without requiring gradient computations or additional training. Extensive experiments on Long-to-Short (L2S) and general merging tasks demonstrate that ACM consistently outperforms all baseline methods. For instance, in the case of Qwen-7B models, TIES-Merging equipped with ACM achieves a **55.3%** reduction in response length while simultaneously improving reasoning accuracy by **1.3** points. Our code is available at ACM

## 1 Introduction

The cognitive evolution of LLMs has progressed from System 1 to System 2 paradigms [16, 46], characterized by the emergence of advanced reasoning models like OpenAI's o1/o3 [29, 30], QwQ [31], and DeepSeek-R1 [6]. System 1 implementations [5, 7], such as GPT-4o [28], LLaMA-3 [7], DeepSeek-V3 [5], leverage rapid intuitive processing for immediate responses but struggle with complex reasoning tasks. In contrast, System 2 architectures are fine-tuned with extended thinking chains to promote deliberate analysis through iterative self-assessment, error mitigation, and verification, albeit facing challenges related to redundancy. This dual-system dichotomy motivates the Long-to-Short (L2S) framework [37], which seeks to reconcile System 2's analytical depth with System 1's operational efficiency. Beyond computationally intensive training-based approaches [1, 10] and instability-prone prompting methods [23], a promising alternative lies in model merging techniques to seamlessly integrate System 2 models with their counterparts without incurring additional computational overhead [40].

39th Conference on Neural Information Processing Systems (NeurIPS 2025).

Model merging [11, 39] refers to the process of integrating the parameters of a pre-trained (PT) model with those of multiple fine-tuned (FT) models to create a single unified model. This approach aims to enhance performance, generalization, and robustness by leveraging the strengths and diverse insights of each individual model. Within the framework of model merging, task vectors [11] have emerged as essential components for encoding task-specific knowledge representations. Defined as parametric deltas between pre-trained and fine-tuned model weights, these vectors theoretically enable cross-model capability integration through linear arithmetic operations. Current task vector-based techniques [42, 45] predominantly use static coefficient protocols that enforce uniform scaling across all parameters and tasks. This simplification assumes uniform task relevance across all neural weights, regardless of their layer-specific functional criticality. However, layer importance is often heterogeneous. In tasks such as L2S, layers critical for specific functions, such as the *lm_head* layer for generation length control, may require different weighting coefficients than other layers. To mitigate the problems, recent efforts utilize gradients [19] or activations [27] to derive layer-specific coefficients for task vectors. Nonetheless, these methods either overlook inter-model relationships by solely focusing on PT model's activations or necessitate complex backpropagation.

In this work, we tackle the aforementioned challenges by theoretically analyzing how activation values relate to weight salience, then designing customized layer-specific weighting coefficients to avoid complex gradient calculations. Mutual information (MI) is selected to measure activations, as it can select features that are highly relevant while preserving critical information, quantifying feature redundancy [8, 14]. Specifically, we propose **A**ctivation-Guided **C**onsensus **M**erging (**ACM**), a novel and efficient approach based on activation MI among models. Different from the previous element-wise activation merging method [27], which concentrates on maintaining the pre-trained model's capacity, our ACM first computes layer-wise activations for both PT and FT models using a **shared** calibration dataset. Next, ACM measures the mutual information (MI) between corresponding PT and task-specific FT activations at each layer, normalizing these values to obtain layer-specific weighting coefficients. The overall framework is presented at Figure 1. Crucially, these coefficients are inversely proportional to the mutual information scores, layers with higher similarity (greater MI) receive lower weights to reduce redundancy, while layers showing significant divergence (lower MI) are assigned higher weights to preserve FT-specific capabilities.

Overall, our contributions include: 1) We analyze the correlation between activation space and weight salience, propose a novel merging approach dubbed as ACM, which leverages MI to compute the weight coefficients of each layer. 2) Extensive experiments in the L2S merging scenario demonstrate ACM's efficiency in maintaining performance and reducing redundancy compared to both training-based and traditional merging baselines. For instance, in the case of Qwen-7B models, TIES-Merging equipped with ACM achieves a **55.3%** reduction in response length while simultaneously improving reasoning accuracy by **1.3** points. 3) General merging tasks further validate ACM's soundness and efficacy. The results show that ACM serves as a plug-and-play solution, consistently improving the performance of existing merging methods.

## 2 Related Work

**Long-to-Short Reasoning**  The paradigm shift in LLM has evolved from rapid, straightforward System 1 reasoning to deliberate, analytical System 2 reasoning frameworks [13, 16, 35]. While System 1 models demonstrate proficiency in generating intuitive responses with low latency, their performance degrades significantly when confronted with complex tasks. In contrast, System 2 models employ recursive self-evaluation mechanisms and error correction protocols to enhance correctness and robustness in multi-step reasoning scenarios. However, this methodological rigor introduces computational inefficiencies, generating excessively verbose responses. Recent advancements in L2S reasoning [37] have emerged as a promising direction to address this fundamental trade-off.

Current research prioritizes two complementary strategies for optimizing reasoning efficiency, namely Chain-of-Thought (CoT) compression and task-adaptive computation allocation. The former approach targets semantic distillation of reasoning paths. For example, TokenSkip [41] implements hierarchical token importance scoring followed by strategic omission, while CoT-Valve [24] optimizes parameter space trajectories through gradient-guided step size modulation. For computational resource allocation, O1-Pruner [23] integrates reinforcement learning (RL) with curriculum-based sampling to prioritize high-complexity tasks, whereas DAST [33] establishes a token-length difficulty metric to dynamically calibrate reward functions. Parallel efforts explore budget-aware RL frameworks

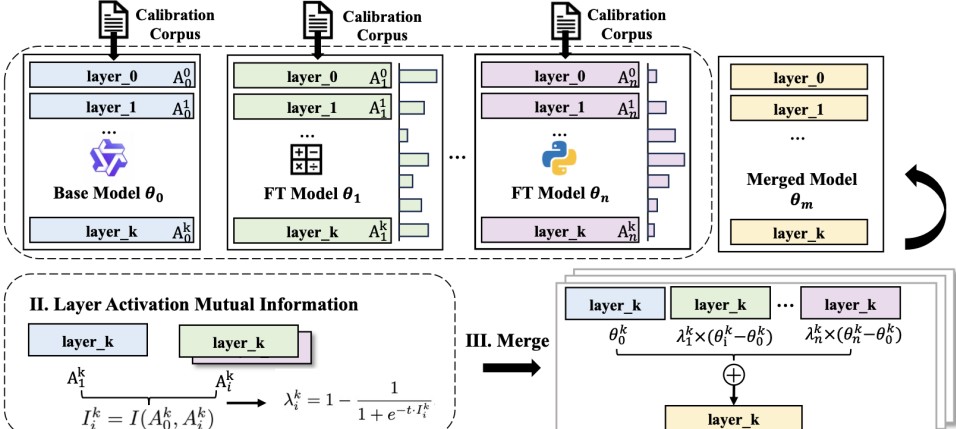

Figure 1: Overall framework of our Activation-Guided Consensus Merging Method, which extracts task-specific layer-wise activation patterns from a shared calibration corpus, quantifies their mutual information with the base model, and performs a weighted synthesis of parameters across models.

for adaptive reasoning length control [1, 10]. Despite progress in prompt engineering techniques [23, 37], like few-shot response truncation and conciseness-oriented demonstration design [26], these methods exhibit critical limitations. Their performance demonstrates pronounced sensitivity to different models and prompts, resulting in inconsistent generalization across task domains.

To alleviate the aforementioned challenges, model merging emerges as a promising strategy to integrate the simplicity of System 1 with the accuracy of System 2, maintaining stable performance without the need for training overhead.

**Model Merging**   Model merging has emerged as a paradigm-shifting alternative to conventional training-based approaches, enabling the integration of multiple task-specific models into a unified one [43]. This technique demonstrates superiority in several scenarios: (1) performance improvement through checkpoints merging [11, 42]; (2) effective mitigation of catastrophic forgetting via parameter-space reconciliation [2], and (3) realization of adaptive long-to-short reasoning capabilities [37].

Task vectors[11] have recently dominated the field of model merging. These methods can be systematically classified into three categories, including 1) arithmetic merging operates fine-tuned features as directional task vectors in parameter space [11, 39, 42, 45]. These approaches perform algebraic operations (e.g., linear combinations or magnitude normalization) to derive consolidated models; 2) low-rank-based merging leverages singular value decomposition (SVD) to identify latent low-rank structures within task vectors [21, 22]. Such methods achieve efficient merging through principal component retention while preserving critical task-specific information, and 3) activation-based merging incorporates input-driven activations [19, 27] and sensitivity-aware balancing mechanisms, thus dynamically adjusting parameter weights based on given context. Parallel to these approaches, Mixture-of-Experts (MoE) based merging strategies [20, 34] constitute a distinct research direction. However, their architectural modifications and model size adjustments place them beyond the scope of parameter-space merging techniques discussed in this work.

Notably, recent works [37, 40] have revealed the exceptional efficacy of model merging on Long-to-Short reasoning tasks, particularly through activation-based methods. Although these approaches [19, 27] demonstrate superior performance across diverse tasks and model scales, they still exhibit significant limitations on the usage of activations, focusing solely on the activations of the pre-trained model, or requiring complex gradient calculations, which motivates our further exploration.

## 3 Methodology

### 3.1 Preliminaries

**Unveiling the Activation-Weight Salience Relation**   Practically, the salience of model parameters is assessed through statistical analysis of activation patterns using a calibration dataset [18, 27, 32].

For a given layer characterized by a weight matrix $W \in \mathbb{R}^{N \times M}$, we analyze its parameter sensitivity by introducing a perturbation matrix $\Delta W \in \mathbb{R}^{N \times M}$, resulting in perturbed weights defined as $W' = W + \Delta W$. For an input vector $\boldsymbol{x} \in \mathbb{R}^N$, the transformation of the perturbed layer can be expressed as $\boldsymbol{y}' = \boldsymbol{x}W' = \boldsymbol{x}W + \boldsymbol{x}\Delta W$.

Let us examine the differential output resulting from weight refinement, defined as $\Delta y := y' - y = \sigma(W'x) - \sigma(Wx)$. By applying Taylor's expansion to the activation function $\sigma$, we derive:

$$\begin{aligned}
\Delta y &= \sigma(W'x) - \sigma(Wx) \\
&= \sigma(Wx) + \langle \nabla\sigma(Wx), W'x - Wx \rangle + O(\|W'x - Wx\|^2) - \sigma(Wx) \\
&= \langle \nabla\sigma(Wx), \Delta Wx \rangle + O(\|\Delta W\|^2) \\
&\approx \langle \nabla\sigma(Wx), \Delta Wx \rangle,
\end{aligned}$$

where the higher-order term $O(\|\Delta W\|^2)$ is negligible due to the sufficiently small magnitude of the weight perturbation.

Based on this approximation, we formally define the weight salience $S(W)$ with respect to calibration input $x$ as follows:

$$S(W; x) := \langle \nabla\sigma(Wx), \Delta Wx \rangle, \quad S(W) := \sup_{\|x\|=1} S(W; x).$$

In the specific case where $\sigma$ is the activation function, we observe that $\nabla\sigma \in \{0, 1\}$. Consequently, the weight salience for the activation function is bounded by $S(W) \leq \sup_{\|x\|=1}\langle 1, \Delta Wx \rangle \leq \|\Delta W\|$. The derived weight salience $S(W)$ reflects activation function sensitivity via gradient-perturbation interactions, revealing how parameter importance emerges from the interplay between weight changes and activation responses.

**Mutual Information**   Mutual Information (MI) is a statistical measure that quantifies the dependency between two random variables, defined as:

$$I(X; Y) = \sum_{x \in X} \sum_{y \in Y} p(x, y) \log \frac{p(x, y)}{p(x)p(y)} \tag{1}$$

This measure indicates how much knowledge of one variable reduces the uncertainty of the other. In feature space, MI assists in selecting features that are highly relevant to the target task, ensuring the preservation of critical information [3, 14, 50]. Additionally, MI quantifies feature redundancy, enabling the optimization of objective functions to minimize this redundancy[8, 48]. The essence of model merging is to integrate complementary information from various models or features, and MI can assist in enhancing merging effectiveness.

In our preliminary attempts, we also considered Kullback-Leibler divergence and cosine similarity to measure model relationships.

- KL divergence is asymmetric; for instance, with $P = [0.9, 0.1]$ and $Q = [0.5, 0.5]$, we found $D_{\mathrm{KL}}(P \parallel Q) \approx 0.368$ and $D_{\mathrm{KL}}(Q \parallel P) \approx 0.511$, leading us to exclude this metric.
- Cosine similarity exhibits substantial variability across layers, approximately ranging from $0.25$ to $0.8$, which can negatively impact performance, as prior studies emphasized that weight coefficient fluctuations should be minimal[19]. Furthermore, the range of cosine similarity is $(-1, 1)$, complicating the interpretation of negative values.

Therefore, We chose MI as a robust tool for examining model relationships.

## 3.2   Activation Consensus Merging

Consider the merging of $N$ specific models with parameters $\theta_1, \theta_2, \ldots, \theta_N$, all derived from a common PT backbone $\theta_0$. The goal of the merged model $\theta_{\mathrm{merged}}$ is to effectively handle multiple tasks simultaneously. Each fine-tuned model's capabilities are encapsulated by task vectors [11], defined as the difference between the task-tuned parameters and the PT backbone: $\delta_i = \theta_i - \theta_0$,   for $i \in \{1, \ldots, N\}$. Based on the task-vector, a single, static merged model is formulated as:

$$\theta_{\text{merged}} = \theta_0 + \frac{1}{N} \sum_{i=1}^{N} \lambda_i \cdot \delta_i$$

As previously noted, due to the differing importance and characteristics of each layer, it is essential to employ distinct weight coefficients rather than a uniform coefficient for all layers. Given a task-agnostic representative calibration corpus $D$, we can feed the calibration data into the models to obtain activations. We denote the layer-wise activations of a model as $A_i^1, A_i^2, \ldots, A_i^k$, where $k$ indicates layer index. Subsequently, we can derive the mutual information between the PT and FT models of each layer across the calibration dataset as:

$$I_i^k = I(A_0^k, A_i^k), \quad k \in K$$

The mutual information values can be relatively large, whereas the range of the weight coefficients is typically constrained between 0 and 1 [42, 45]. To ensure consistency, we employ the sigmoid function for normalization, as described below:

$$\lambda_i^k = 1 - \frac{1}{1 + e^{-t \cdot I_i^k}}, \tag{2}$$

where $t$ is the hyperparameter, practically ranging from -1 to 1, and $\lambda_i^k$ represents the layer-specific importance of each task vector. A lower coefficient indicates higher mutual information, suggesting significant overlap between the two models. In the context of L2S tasks, this overlap may correlate with shared fundamental generative capabilities, thereby justifying smaller weight coefficients. Conversely, a higher weight coefficient may signify that those particular weights play a more critical role in enhancing reflective capabilities.

Overall, the merged layer $k$ for the model $\theta_{\text{merged}}$ is expressed as:

$$\theta_{\text{merged}}^k = \theta_0^k + \frac{1}{N} \sum_{i=1}^{N} \lambda_i^k \cdot \delta_i^k, \quad k \in K \tag{3}$$

We perform the above operations on each layer to obtain the merged model. The overall framework can be found in Figure 1. We would like to highlight that our method is plug-and-play, aiming to identify the optimal weight coefficients $\lambda_i^k$. These coefficients can be seamlessly integrated into various existing task-vector-based merging methods, such as TA [11] and TIES-Merging [42].

## 4 Experiments

We evaluate our method on both Long-to-Short merging tasks and general model merging tasks. The results show that our approach consistently outperforms existing merging strategies across diverse benchmarks, demonstrating stronger cross-task adaptability and knowledge integration capabilities.

### 4.1 Long-to-Short Merging

**Models and Datasets** Long-to-Short Merging aims to combine the strengths of fast (System 1) and slow-thinking (System 2) models, achieving high accuracy with reduced redundancy. To evaluate, we conducted experiments across various model sizes and tasks, utilizing Qwen2.5-Math (1.5B, 7B), Qwen2.5 (14B, 32B), and their corresponding DeepSeek-R1 distilled series models [6]. Model performance is assessed on established reasoning datasets, including GSM8K [4], MATH500 [17], Minerva Math [15], Olympiadbench [9], College Math [36], and AIME 2024[1]. To ensure reproducibility, we employed the public evaluation toolkit provided by QwenLM [2], adhering to their recommended versions of dependencies. Code generation capabilities are measured on HumanEval-Pro [47] and LiveCodeBench [12]. Following DeepSeek-AI et al. [6], fast-thinking models are evaluated in a few-shot context, while slow-thinking models are in a zero-shot framework.

---

[1]https://huggingface.co/datasets/Maxwell-Jia/AIME_2024
[2]https://github.com/QwenLM/Qwen2.5-Math

Table 1: Evaluations of different model merging methods on Qwen-7B models. The number in `()` indicates the average response length on the dataset. Length reduction comparisons are conducted with R1, the superior results of the merging methods on each benchmark are highlighted in bold.

| Method \ Bench | GSM8K | MATH500 | Minerva Math | Olympiad Bench | College Math | AIME24 | Avg. |
|---|---|---|---|---|---|---|---|
| Qwen2.5-Math-7B | 88.9 | 52.2 | 12.1 | 17.5 | 22.6 | 3.3 | 32.8 |
| | (130.2) | (526.2) | (956.7) | (1259.2) | (794.8) | (1528.0) | |
| DeepSeek-R1-7B | 89.3 | 87.4 | 36.4 | 51.0 | 39.8 | 23.3 | 54.5 |
| | (1062.0) | (2825.9) | (3055.9) | (5793.8) | (2461.6) | (8675.4) | |
| DPO | 89.6 | 88.4 | 33.8 | 48.6 | 39.7 | 40.0 | 56.7 |
| | (983.6) | (2587.1) | (2304.9) | (4932.3) | (2073.5) | (7029.5) | (-15.0%) |
| *Arithmetic Merging Methods* | | | | | | | |
| Average Merging | 81.6 | 78.2 | 30.9 | 37.9 | 36.1 | 26.7 | 48.6 |
| | (636.4) | (1416.7) | (2277.4) | (3202.2) | (2065.7) | (5964.8) | (-34.6%) |
| Task Arithmetic | 90.5 | 83.4 | **41.9** | 45.0 | 40.3 | 20.0 | 53.5 |
| | (617.5) | (1617.4) | (1416.2) | (2650.6) | (1443.9) | (3594.6) | (-48.7%) |
| TIES-Merging | 90.6 | 81.8 | 38.2 | 43.0 | 41.9 | **33.3** | 54.8 |
| | (552.2) | (1492.9) | (1349.2) | (2473.7) | (1287.8) | (3302.1) | (-53.0%) |
| DARE | 84.0 | 75.4 | 29.4 | 35.7 | 36.2 | 23.3 | 47.3 |
| | (815.6) | (2237.1) | (2317.8) | (3266.3) | (2072.1) | (3803.1) | (-30.6%) |
| DARE-TA | 87.9 | **84.0** | 29.4 | 41.6 | 38.3 | 26.7 | 51.3 |
| | (600.4) | (1703.9) | (1567.2) | (2774.8) | (1533.9) | (3454.2) | (-47.0%) |
| DARE-TIES | 89.6 | 82.4 | 37.5 | 41.3 | 40.2 | 23.3 | 52.4 |
| | (584.3) | (1589.4) | (1307.4) | (2655.5) | (1378.5) | (3579.9) | (-50.4%) |
| *Activation-based Merging Methods* | | | | | | | |
| AIM | 90.8 | 83.0 | 40.8 | 46.4 | **42.3** | 26.7 | 55.0 |
| | (540.6) | (1374.5) | (1229.8) | (2323.8) | (1249.6) | (3265.9) | (*-55.3%*) |
| Sens-Merging | 91.2 | 83.4 | 41.5 | 43.9 | 40.2 | 30.0 | 55.0 |
| | (604.2) | (1613.3) | (1402.1) | (2673.0) | (1454.0) | (3355.8) | (-49.4%) |
| ACM-TA | 90.6 | 83.8 | 38.6 | **46.7** | 40.1 | **33.3** | 55.5 (+1.0) |
| | (652.4) | (1636.1) | (1514.1) | (2706.2) | (1494.5) | (3409.9) | (-47.4%) |
| ACM-TIES | **92.2** | **84.0** | 38.6 | 46.4 | 40.3 | **33.3** | **55.8(+1.3)** |
| | (538.3) | (1450.1) | (1153.1) | (2356.8) | (1284.6) | (3076.4) | (*-55.3%*) |

For activation-based merging, the **s1K** dataset [25] is used as the calibration data, which provides aligned short- and long-CoT answers for each question. The dataset containing about 1000 pieces of data is first clustered by the K-means algorithm into 20 categories, followed by an even sampling of 10% of the total data. To enhance numerical stability, we perform a translation transformation by subtracting the maximum MI obtained from different data pieces. The maximum sequence lengths for quick and slow-thinking models are set to 8K and 10K, respectively. All models are uploaded and evaluated with the BF16 data type. We report the average scores across five runs with different random seeds.

**Baselines** We evaluate the effectiveness of our ACM by comparing it with individual task-specific models and established model-merging techniques, including Average Merging, Task Arithmetic (TA) [11], TIES-Merging [42], DARE [45], AIM [27], and Sens-Merging [19]. TA enhances the merging process by introducing task vectors, demonstrating that simple arithmetic operations on these vectors can effectively generate a merged output. DARE and TIES utilize pruning-then-scaling methods to merge these vectors, based on the premise that not all parameters contribute equally to overall performance. AIM and Sens-Merging are designed to be plug-and-play, allowing them to integrate seamlessly with the existing methods by assigning layers with different importance weights. AIM computes only the PT model's activations, overlooking the FT model's task characteristics. Sens-Merging requires complex backpropagation to calculate sensitivity.

**Hyperparameters** Both the baseline methods and our ACM-enhanced methods use the same range of weight coefficients for fair comparison. When applying Task Arithmetic, we utilize a scaling coefficient of $\lambda = 0.7$ by default. This choice ensures that the task vector's original magnitude

Table 2: Evaluations of various model merging methods on Qwen-1.5B models.

| Method \ Bench | GSM8K | MATH500 | Minerva Math | Olympaid Bench | College Math | AIME24 | Avg. |
|---|---|---|---|---|---|---|---|
| Qwen2.5-Math 1.5B | 75.9 | 36.2 | 11.4 | 22.8 | 5.6 | 0.0 | 25.3 |
| | (118.1) | (411.0) | (1036.8) | (607.7) | (864.9) | (864.9) | |
| DeepSeek-R1 1.5B | 76.6 | 69.6 | 15.1 | 30.4 | 34.2 | 20.0 | 41.0 |
| | (2743.3) | (4508.2) | (6374.2) | (7389.3) | (4058.1) | (8952.3) | |
| Average Merging | 26.6 | 12.2 | 7.4 | 5.6 | 4.9 | 0.0 | 9.5 |
| | (3248.7) | (3890.6) | (3706.0) | (3928.6) | (3853.4) | (3977.3) | (-24.11%) |
| Task Arithmetic | 74.5 | 62.6 | 21.0 | 28.6 | 28.9 | 10.0 | 37.6 |
| | (549.7) | (1619.4) | (1671.0) | (2588.6) | (1321.7) | (3395.3) | (-68.7%) |
| TIES-Merging | 75.7 | 66.2 | 22.1 | 30.8 | 29.6 | 10.0 | 39.1 |
| | (414.0) | (1038.5) | (1066.8) | (1963.7) | (876.7) | (2901.0) | **(-75.5%)** |
| AIM | 38.7 | 35.4 | 11.4 | 22.8 | 5.6 | **16.7** | 21.8 |
| | (1084.9) | (2034.7) | (2510.9) | (3301.9) | (1464.6) | (2105.1) | (-61.9%) |
| Sens-Merging | 71.3 | 63.8 | 24.8 | 28.9 | 30.5 | 13.3 | 38.8 |
| | (720.0) | (1611.3) | (1774.1) | (2683.8) | (1394.7) | (3546.7) | (-66.7%) |
| ACM-TA | 76.8 | 68.8 | 25.3 | 31.1 | 29.6 | **16.7** | 41.4(+0.4) |
| | (438.0) | (1309.6) | (1214.7) | (2291.6) | (1067.3) | (3240.9) | (-73.7%) |
| ACM-TIES | **78.4** | **71.4** | **28.7** | **33.8** | **37.9** | 10.0 | **43.3(+2.3)** |
| | (962.1) | (1235.6) | (1350.5) | (1849.6) | (848.5) | (2688.5) | (-73.5%) |

is maintained when integrated with the pretrained backbone. In contrast, the DARE technique is more susceptible to changes in both the scaling coefficient, $\lambda$, and the drop rate, $r$. Consequently, to obtain stable performance with DARE, we configure its scaling coefficient to $\lambda = 0.7$ and its default drop rate to $r = 0.3$. For AIM, we utilize $\omega = 0.4$ as recommended. As for Sens-Merging and TIES-Merging, we consistently follow settings reported by previous work [40] for different models. For our ACM, we set $t = 0.7$.

**Main Results** As presented in Tables 1 and 2, we provide the results for the 1.5B and 7B models. For comprehensive results regarding the code benchmark and experimental details for the 14B and 32B models, please refer to Appendix A. The following observations can be derived:

**(1) ACM exhibits substantial improvements in accuracy across various model scales.** For the 7B model (Table 1), ACM-TA and ACM-TIES achieve average accuracies of **55.5%** and **55.8%**, respectively, outperforming both task vector merging methods (e.g., Task Arithmetic at 53.5%) and activation-based approaches like AIM (55.0%). With the 1.5B model (Table 2), ACM-TA and ACM-TIES attain accuracies of **41.4%** and **43.3%**, significantly surpassing TIES-Merging (39.1%) and Sens-Merging (38.8%). Experiments on larger models (14B and 32B) further confirm ACM's effectiveness, with integrated TA and TIES strategies consistently enhancing performance. These results underscore the necessity of establishing appropriate layer-specific weight coefficients. ACM systematically derives these coefficients for each layer by analyzing the mutual information relationships among the models, thereby enhancing performance.

**(2) ACM significantly mitigates redundant outputs while enhancing inference efficiency in smaller-scale models.** Figure 6 demonstrates that ACM-TA and ACM-TIES achieve sequence lengths of approximately 1,000 tokens in the 1.5B code generation task, significantly shorter than the 6,000+ tokens required by the DeepSeek counterpart. Similar trends are observed for the 14B code task (Table 8). In mathematical reasoning tasks, ACM-TIES reduces sequence length by **55.3%** for the 7B model (Table 1) and **73.5%** for the 1.5B model (Table 2), highlighting ACM's efficiency in compressing outputs without compromising performance.

Specifically, consistent with previous findings [40], we observed that response length positively correlates with question difficulty. Furthermore, as shown in the appendix A.3, the merged model retains reflective capabilities; however, reflection frequency has decreased due to the PT fast-thinking model. According to our careful case study, while the merged model maintains favorable reasoning ability, it avoids redundant reflection on simpler mathematical problems, such as those in GSM8K, thereby reducing response length.

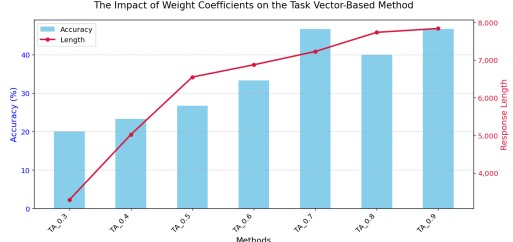

Figure 2: The impact of weight coefficients of the task vector-based merging on Qwen-14B models. As weight coefficients increase, accuracy improves while response length grows.

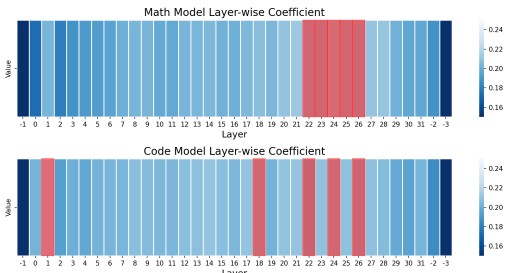

Figure 3: Layer-wise coefficients across different task-specific models, with the Top-5 coefficients highlighted in red.

Table 3: Evaluation of various methods integrating three LLaMA series models. Our ACM demonstrates an improvement combined with other approaches. Comparisons are conducted with baseline merging methods, the superior results among them on each benchmark are highlighted in bold.

| Bench Method | GSM8K | HellaSwag | MBPP-Pro | Avg. |
|---|---|---|---|---|
| Llama-2-7b-hf | 16.60 | 69.01 | 3.17 | 29.59 |
| MammoMATH | 46.45 | 67.09 | 11.10 | 41.55 |
| codellama-7b | 17.13 | 30.18 | 29.10 | 25.47 |
| TA | 23.58 | 69.05 | 6.61 | 33.08 |
| TIES | 22.90 | 68.98 | 8.99 | 33.62 |
| ACM-TA | **23.68** (+1.0) | **69.27** (+0.22) | 6.55 (-0.06) | 33.18 (+0.1) |
| ACM-TIES | 22.82 (-0.08) | 69.16 (+0.18) | **9.11** (+0.12) | 33.70 (+0.08) |

**(3) ACM highlights the necessity of layer-specific coefficients in model merging.** As depicted in Figure 4, the weight coefficients between the *lm_head* layer and the *embed* layer in the L2S task are considerably lower than those of the intermediate layers, with the mutual information in the former being around 4, whereas the mutual information in the intermediate layers is approximately 2.5. This discrepancy may stem from the knowledge distillation process between the slow- and fast-thinking models: shared pretrained semantic representations at these layers necessitate only minimal parameter adjustments during alignment. Notably, this pattern persists consistently across models of varying scales, underscoring the robustness of the observed behavior.

**(4) Balancing accuracy improvement and length reduction in larger-scale models proves challenging.** Experiments show that in large models like the 14B and 32B, setting the task vector coefficient around 0.7 improves accuracy but difficulty achieving notable length reduction as smaller models, sometimes even exceeding that of the slow-thinking model, aligning with the findings in Wu et al. [40]. Moreover, Figure 2 explores that higher weight coefficients enhance slow-thinking features, resulting in longer outputs and improved accuracy. Smaller models, however, can gain accuracy while maintaining shorter lengths. This discrepancy arises from the significant parameter redundancy in large models, which enables the coexistence of both fast and slow-thinking modes, complicating the process of length reduction.

Further analysis highlights that task characteristics affect the accuracy-length trade-off. As shown in Table 8, the 14B model compresses output length while enhancing accuracy in coding tasks, due to strict syntactic constraints and clear logical goals. This enables effective elimination of redundancy while ensuring semantic correctness. In contrast, reasoning tasks are more prone to conflicts between accuracy and length due to a lack of structural constraints.

## 4.2 General Model Merging

**Models and Datasets** We further validate our ACM approach on general model merging tasks, ensuring that the merged model retains the capabilities of the baseline models. In addition to Qwen, we also apply our ACM to the LLaMA architecture by merging multiple LLaMA-based FT models and compare against strong baselines, including Llama-2-7b-hf [38], MammoMATH[3], a fine-tuned

---

[3]huggingface.co/TIGER-Lab/MAmmoTH-7B

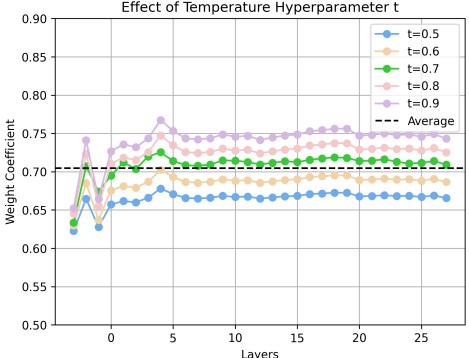

Figure 4: Ablation study of hyperparameter $t$ on 7B L2S task. "-1" layer corresponds to the *embed* layer, "-2" layer represents the *model.norm* layer, and "-3" layer implies the *lm_head* layer.

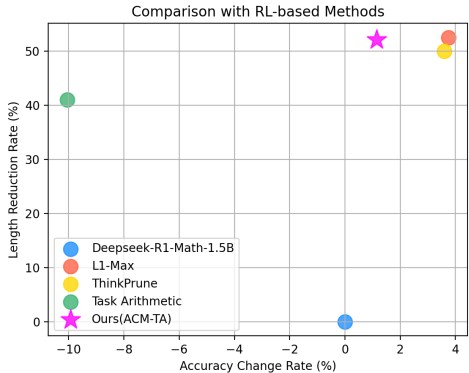

Figure 5: Comparison with training-based methods regarding the trade-off between response length and accuracy.

version for mathematical reasoning, and CodeLlama-7b-hf[4], a code-specialized model designed for general code synthesis and understanding. We assess performance on GSM8K, a dataset of grade school math word problems; HellaSwag [49], a commonsense reasoning benchmark; and MBPP-Pro [47], a dataset of Python code generation problems.

**Main Results**   To ensure the merged model retains the capabilities of all individual models, we set the hyperparameter for the TA method to 0.2, while the TIES method uses coefficients of 0.7 and 0.2, respectively. Accordingly, the hyperparameter $t$ for our ACM method is set to -1.8. Table 3 illustrates the effectiveness of the ACM method when integrated with existing approaches to enhance the performance of LLaMA-based models across three benchmark tasks. The baseline models exhibit distinct performance characteristics on these benchmarks. When combined with the TA and TIES methods, the ACM method demonstrates varying degrees of improvement. Notably, the merging methods yield limited performance on the MBPP-Pro and GSM8K tasks compared to the baseline, likely due to the low TA coefficient and the relatively inferior performance of two of the three models. In contrast, both ACM-TA and ACM-TIES achieve more consistent improvements on HellaSwag, where the baseline models align more closely. The incorporation of ACM enhances robustness and generalization capabilities, particularly in scenarios with limited model diversity and strategically optimized weight allocation. These findings highlight ACM's feasibility as a versatile and effective strategy for synergistically combining multiple large language models across various domains.

Additionally, Figure 3 also demonstrates that different models and tasks require distinct layer-wise coefficients for optimal performance. The uneven distribution of coefficients across layers, along with the task-specific focus on certain layers, emphasizes the necessity for adaptive weighting strategies.

## 5   Further Analyses

**Ablation Study on Hyperparameter**   To investigate the sensitivity of our model merging framework to the hyperparameter $t$, we conducted an ablation study analyzing its impact on layer-wise weight coefficients, as shown in Figure 4. The results demonstrate that varying $t$ within the range [0.5, 0.9] induces only minor fluctuations in the learned weights across all layers. This highlights the robustness of our adaptive weighting mechanism to hyperparameter choices. Moreover, our findings indicate that across experiments involving models ranging from 1.5B to 32B, the ACM method consistently yielded favorable outcomes with a fixed hyperparameter 0.7. In contrast, the Sens-Merging method displayed significant variability, with its temperature coefficient fluctuating between 1 and 6, while the scaling coefficient varied from 0.4 to 0.8. A similar trend was observed in the TIES-Merging method. Collectively, these results suggest that the ACM method exhibits superior robustness, both in terms of the number of hyperparameters and the breadth of parameter adjustments.

---

[4]https://huggingface.co/codellama/CodeLlama-7b-hf

Table 4: Effect of the number of calibration data.

| Pieces | 20 | 50 | 100 | 200 | 300 |
|---|---|---|---|---|---|
| Acc. | 91.4 | 90.8 | 92.2 | 91.8 | 92.1 |
| Length | 623.63 | 624.17 | 538.3 | 585.65 | 603.0 |

Table 5: Effect of different random seeds.

| | seed a | seed b | seed c |
|---|---|---|---|
| Acc. | 91.9 | 91.7 | 92.4 |
| Length | 558.33 | 553.58 | 541.41 |

Table 6: Effect of different clusters.

| Clusters | random | 10 | 20 | 30 |
|---|---|---|---|---|
| Acc. | 88.6 | 91.1 | 91.8 | 91.6 |
| Length | 643.4 | 602.2 | 538.3 | 586.7 |

Table 7: Performance on IFEval.

| | prompt-level | instruction-level |
|---|---|---|
| TIES | 18.48 | 30.58 |
| ACM-TIES | 20.15 (+1.67) | 31.06 (+0.48) |

**Ablation Study on Calibration Dataset** We evaluate the robustness of **ACM** from multiple dimensions, including the number of sampled calibration data pieces, the number of clusters employed prior to sampling, the choice of random seeds, and the use of distinct calibration datasets. The Experiments are conducted on Qwen2.5-Math-1.5B model series. Tabel 4 indicates that once the number of calibration data points reaches 100, accuracy remains high while length remains short. We also analyzed the weight coefficients at this threshold. When the number of data is below 100, the distribution of weight coefficients fluctuates significantly, with the top five layers showing considerable variation. Conversely, when the count exceeds 100, these characteristics stabilize. This trend is similar across models such as 1.5B and 14B. Thus, we advocate for a 10% sampling rate in our experimental design. We also present the results of the 7B model using different sampling seeds in Table 5, It is evident that our clustering and sampling methods are both reasonable and robust. Additionally, we investigated the effects of varying the number of clusters. Table 6 demonstrates that random clustering results in suboptimal performance, indicating possible imbalance in the dataset's distribution. Notably, with and beyond 20 clusters, we observed improved accuracy and reduced length, suggesting that clustering then sampling effectively mitigates data imbalance. We further validate the effectiveness of **ACM** using calibration datasets other than S1K; detailed experimental results are provided in the Appendix A.4.

**Broader Task** We additionally selected the IFEval dataset [51] —a benchmark for instruction following—to evaluate our method. For simplicity, we use 1.5B Qwen model and choose TIES as our baseline. The experimental results are shown in Table 7 We find that on the instruction-following dataset, **ACM** can further enhance the model merging performance of TIES, thereby further validating the effectiveness of our **ACM** approach.

**Comparison with RL-based Methods** Figure 5 presents a comparative analysis of our proposed method (ACM-TA) against both the baseline model and RL-based training methods in terms of accuracy-length trade-off. Our approach demonstrates three key advantages: 1) It not only improves Deepseek-R1's accuracy by approximately 1.2%, but also achieves about 51% length reduction rate; 2) Despite requiring no training cost, ACM-TA matches the length compression efficiency of training-intensive methods like L1-Max [1] (about 52%) and ThinkPrune (about 50%) [10], though the accuracy improvement is relatively limited compared to training-based methods; 3) Compared to its non-training counterpart Task Arithmetic (TA), ACM-TA turn TA's 10% accuracy drop at 40% compression to slight performance improvement at 50% compression. This highlights our method's unique ability to simultaneously enhance model accuracy and efficiency through adaptive merging. Besides, our training-free **ACM** method requires only tens of seconds to one or two minutes (depending on model size), exhibiting extremely low time overhead and high efficiency.

# 6 Conclusion

We propose **ACM**, a novel and efficient activation-based model merging method, which analyzes the MI of activations between models to calculate appropriate weight coefficients. Experimentas on L2S reasoning tasks and general merging tasks demonstrate the effectiveness of ACM. Further comparison with RL-based approaches indicates that our method can achieve comparable length reduction on the L2S task without training. We believe our work can inspire future research in this field.

## Acknowledgments

This work was supported in part by the Research Grants Council of the Hong Kong SAR under Grant Collaborative Research Fund C1042-23GF and GRF 11217823, the National Natural Science Foundation of China under Grant 62371411, InnoHK initiative, the Government of the HKSAR, Laboratory for AI-Powered Financial Technologies.

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

Table 8: Evaluations of various model merging methods on Qwen-14B models.

| Bench / Method | GSM8K | MATH500 | AIME24 | HuamnEval Pro | Avg. |
|---|---|---|---|---|---|
| Qwen2.5-14B | 90.8 | 47.2 | 3.3 | 51.8 | 48.3 |
| | (138.5) | (522.9) | (1493.8) | (101.4) | |
| DeepSeek-R1-14B | 91.9 | 89.0 | 50.0 | 61.0 | 73.0 |
| | (558.8) | (2314.1) | (7724.8) | (229.5) | |
| Average Merging | 91.5 | 88.2 | 46.7 | 63.4 | 72.5 |
| | (642.8) | (2925.1) | (7922.7) | (158.0) | (+3.2%) |
| Task Arithmetic | 94.1 | 85.4 | 33.3 | 63.4 | 69.1 |
| | (565.9) | (2191.4) | (6873.6) | (160.1) | (-11.3%) |
| TIES-Merging | 93.4 | 84.6 | 26.7 | 62.8 | 66.9 |
| | (503.1) | (2169.7) | (6953.7) | (92.0) | (-21.55%) |
| Sens-Merging | **94.8** | 86.2 | 46.7 | 63.4 | 72.8 |
| | (641.0) | (2697.5) | (7155.4) | (132.8) | (-4.58%) |
| ACM-TA | 94.2 | 87.8 | 46.7 | **64.6** | 73.3 (+0.3) |
| | (608.9) | (2650.4) | (7376.6) | (159.1) | (-2.93%) |
| ACM-Average | 92.6 | **88.6** | **50.0** | 64.1 | **73.8(+0.8)** |
| | (591.4) | (2981.4) | (7971.8) | (160.4) | (+1.92%) |

# A  Technical Appendices and Supplementary Material

## A.1  Code Evaluation on 1.5B Models

The experimental results in Figure 6 demonstrate the effectiveness of our ACM-based methods (ACM-TA and ACM-TIES) on the LiveCode benchmark. Compared to the baseline TA method, ACM-TA achieves a 0.7% accuracy improvement while reducing average response length by 12%. Similarly, ACM-TIES outperforms the TIES method by 0.2% accuracy and a 9% shorter response length, showcasing the benefits of integrating activation-aware calibration into existing frameworks. These gains highlight ACM's ability to enhance model performance without sacrificing computational efficiency, particularly in long-context code generation tasks.

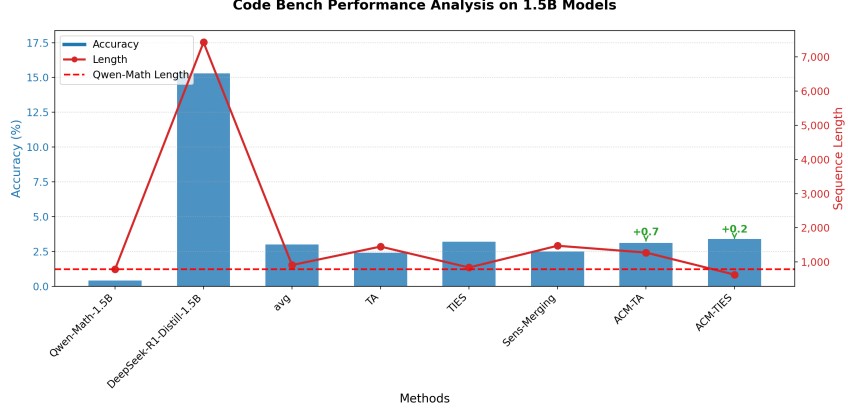

Figure 6: Performance comparison of different methods on LiveCode Benchmark

## A.2  Experiments on Larger-scale Models

The results in Table 8 demonstrate the superiority of ACM-based methods across multiple benchmarks. ACM-TA achieves a 4.2% higher average accuracy than Task Arithmetic while maintaining comparable efficiency, as indicated by similar computational costs (e.g., GSM8K: 608.9 vs. 565.9). Similarly, ACM-Average outperforms Average Merging by 1.3% with slightly increased resource

Table 9: Evaluations of various model merging methods on Qwen-32B models.

| Method \ Bench | GSM8K | MATH500 | AIME24 | Avg. |
|---|---|---|---|---|
| Qwen2.5-32B | 92.3 | 55.4 | 6.7 | 51.5 |
| | (130.3) | (497.2) | (1127.9) | |
| DeepSeek-R1-32B | 95.7 | 89.2 | 60.0 | 81.6 |
| | (822.6) | (2621.8) | (7194.2) | |
| Task Arithmetic | 95.1 | 88.8 | **46.7** | 76.9 |
| | (586.7) | (2171.2) | (6752.3) | (-17.3%) |
| TIES-Merging | 95.4 | 87.8 | 43.3 | 75.5 |
| | (516.9) | (1898.7) | (5823.3) | (-27.9%) |
| ACM-TA | **95.6** | **89.8** | **46.7** | **77.4 (-4.2)** |
| | (544.8) | (2124.4) | (6735.3) | (-19.7%) |

usage. Notably, ACM methods consistently achieve top-tier performance on challenging benchmarks like HumanEval Pro (64.6%) and AIME24 (46.7%), surpassing all other merging strategies.

Likewise, the experimental results in Table 9 also demonstrate the superiority of the ACM-TA method on Qwen-32B models. Compared to Task Arithmetic, ACM-TA achieves a 0.5% higher average accuracy (77.4% vs. 76.9%) while maintaining comparable computational efficiency, as evidenced by similar sequence length reductions (e.g., -11.7% vs. -10.7%).

## A.3 Reflection Statistics

Table 10, and 11 present the number of reflections using different merging methods across models of varying scales on different benchmarks. We determine whether reflection has occurred based on the presence of keywords, including: "wait, re-examine, recap, double-check, let me (just) check, and let me (just) verify".

Table 10: Number of responses containing reflective content across various datasets on 1.5B models.

| Method \ Bench | GSM8K (1319) | MATH500 (500) | Minerva Math (272) | Olympiad Bench (675) | College Math (2818) | AIME24 (30) | Avg. |
|---|---|---|---|---|---|---|---|
| Qwen2.5-1.5B | 0 | 0 | 0 | 0 | 0 | 0 | 0 |
| DeepSeek-R1-1.5B | 1315 | 499 | 234 | 668 | 2769 | 30 | 914.2 |
| Average Merging | 774 | 306 | 173 | 369 | 1892 | 16 | 588.3 |
| Task Arithmetic | 151 | 187 | 82 | 295 | 818 | 21 | 259.0 |
| TIES-Merging | 22 | 45 | 13 | 85 | 118 | 3 | 47.7 |
| Sens-Merging | 211 | 205 | 87 | 304 | 909 | 15 | 288.5 |
| ACM-TA | 63 | 107 | 33 | 175 | 353 | 13 | 124.0 |
| ACM-TIES | 137 | 166 | 60 | 280 | 722 | 18 | 230.5 |

## A.4 Different Calibration Dataset

We have also chosen the recently released LIMO [44] dataset as the calibration dataset for our experiments. LIMO challenges conventional wisdom in mathematical reasoning by showing that models can achieve superior performance with significantly less, yet higher quality, training data. In line with the settings applied in s1K, we conducted experiments using LIMO on GSM8K and Minerva Math, and the results are presented in Table 12. Our method demonstrates good adaptability to the new calibration dataset and exhibits robustness.

## A.5 Case Study of ACM

Table 11: Number of responses containing reflective content across various datasets on 14B models.

| Bench Method | GSM8K (1319) | MATH500 (500) | AIME24 (30) | Avg. |
|---|---|---|---|---|
| Qwen2.5-14B | 2 | 0 | 0 | 0.7 |
| DeepSeek-R1-14B | 678 | 473 | 30 | 393.7 |
| Average Merging | 372 | 457 | 30 | 286.3 |
| Task Arithmetic | 861 | 572 | 29 | 487.3 |
| TIES-Merging | 991 | 455 | 29 | 491.7 |
| AIM | 568 | 462 | 30 | 353.3 |
| Sens-Merging | 714 | 483 | 29 | 408.7 |
| ACM-TA | 717 | 481 | 27 | 408.3 |
| ACM-Average | 329 | 451 | 29 | 269.7 |

Table 12: Effect of Different Calibration Dataset

| | GSM8K: Acc. (Length) | Minerva Math: Acc. (Length) |
|---|---|---|
| Qwen-Math-1.5B | 75.9 (118.1) | 11.4 (1036.8) |
| DeepSeek-R1-1.5B | 76.6 (2743.3) | 15.1 (6374.2) |
| Task Arithmetic | 74.5 (549.7) | 21.0 (1671.0) |
| ACM-TA-s1K | 76.8 (438.0) | 25.3 (1214.7) |
| ACM-TA-LIMO | 76.9 (604.9) | 22.3 (1549.7) |

Table 13: Case study of ACM and other merging methods on GSM8K

| | |
|---|---|
| **Question** | Garrett is popping popcorn for a snack. As the pan of kernels heats up, the kernels start popping faster. Twenty pop in the first 30 seconds of cooking, then three times that amount in the next 30 seconds. The kernels increase to four times the initial popping rate in the next thirty seconds, but in the final 30 seconds, the popping slows down to half the rate as the past 30 seconds. After Garrett takes the pan off the heat, a quarter of the number of kernels that popped in the final 30 seconds of cooking also pop from the residual heat. How many pieces of popcorn does Garrett have to eat? |
| **Gold Answer** | 210 |
| **Qwen2.5** | Let's think step by step. In the first 30 seconds, 20 kernels pop. In the next 30 seconds, 3 times that amount pop, so 20 * 3 = 60 kernels pop. In the next 30 seconds, 4 times the initial popping rate occurs, so 20 * 4 = 80 kernels pop. In the final 30 seconds, the popping slows down to half the rate as the past 30 seconds, so 60 / 2 = 30 kernels pop. In the final 30 seconds, a quarter of the number of kernels that popped in the final 30 seconds also pop from the residual heat, so 30 / 4 = 7.5 kernels pop. Since we can't have half a kernel, we'll round down to 7 kernels popping from the residual heat. In total, Garrett has 20 + 60 + 80 + 30 + 7 = 207 pieces of popcorn to eat. The answer is 207 . |

**DeepSeek-R1**

Okay, let's try to figure out how many pieces of popcorn Garrett has to eat.
So, he's popping popcorn in four different 30-second intervals,
and the popping rate changes each time.
Then, after taking the pan off the heat, some more kernels pop from the residual heat.
Hmm, that sounds a bit complicated, but let's break it down step by step.
First, let's list out the information given:
1. In the first 30 seconds, 20 kernels pop.
2. In the next 30 seconds, three times that amount pop. So, that's 3 * 20 = 60 kernels.
3. In the next 30 seconds, the popping rate increases to four times the initial rate.
The initial rate was 20 kernels per 30 seconds,
so four times that would be 4 * 20 = 80 kernels.
4. In the final 30 seconds, the popping slows down to half the rate of the previous interval.
The previous interval was 80 kernels, so half of that is 40 kernels.
5. After taking the pan off the heat, a quarter of the number of kernels that
popped in the final 30 seconds also pop from the residual heat.
So, that's a quarter of 40, which is 10 kernels.
Alright, so let's add up all these numbers to find the total number of kernels popped
during cooking and then add the ones that popped from residual heat.
First interval: 20 kernels.
Second interval: 60 kernels.
Third interval: 80 kernels.
Fourth interval: 40 kernels.
Residual heat: 10 kernels.
So, total kernels popped during cooking: 20 + 60 + 80 + 40.
Let me calculate that: 20 + 60 is 80, plus 80 is 160, plus 40 is 200.
Then, residual heat adds another 10 kernels.
So, total kernels: 200 + 10 = 210.
Wait, but let me double-check to make sure I didn't miss anything.
First 30 seconds: 20.
Next 30 seconds: 3 times 20 is 60. So, 20 + 60 = 80.
Next 30 seconds was 4 times the initial rate, which was 20, so 4 * 20 = 80. So, 80 + 80 = 160.
Final 30 seconds: half the previous rate, which was 80, so half is 40. So, 160 + 40 = 200.
Then, residual heat: a quarter of the final 30 seconds,
which was 40, so 40 / 4 = 10. So, 200 + 10 = 210.
Hmm, that seems consistent. So, the total number of popcorn kernels is 210.
Wait, but let me make sure I interpreted the rates correctly. The problem says:
"Twenty pop in the first 30 seconds of cooking,
then three times that amount in the next 30 seconds.
The kernels increase to four times the initial popping rate in the next thirty seconds,
but in the final 30 seconds, the popping slows down to half the rate as the past 30 seconds."
So, initial rate is 20 per 30 seconds.
Next interval: 3 times that, so 60.
Next interval: 4 times the initial rate, which is 4 * 20 = 80.
Final interval: half the rate of the past 30 seconds, which was 80, so half is 40.
Yes, that seems right.
Then, residual heat: a quarter of the final 30 seconds,
which was 40, so 10.
So, total is 20 + 60 + 80 + 40 + 10 = 210.
I think that's correct. So, the answer should be 210.
*Final Answer**
Garrett has to eat \boxed{210} pieces of popcorn.
</think>
Garrett is popping popcorn in four different 30-second intervals,
with the popping rate changing each time.
After taking the pan off the heat, some additional kernels pop from the residual heat.
Here is the breakdown of the popping rates and the total number of kernels:
1. In the first 30 seconds, 20 kernels pop.
2. In the next 30 seconds, three times that amount pop: (3 times 20 = 60) kernels.
3. In the next 30 seconds, the popping rate increases to four times the initial rate:
(4 times 20 = 80) kernels.
4. In the final 30 seconds,
the popping slows down to half the rate of the previous interval: ($\frac{80}{2} = 40$) kernels.
5. After taking the pan off the heat, a quarter of the number of kernels
that popped in the final 30 seconds also pop from the residual heat: ($\frac{40}{4} = 10$) kernels.
Adding up all these numbers:
- First interval: 20 kernels
- Second interval: 60 kernels
- Third interval: 80 kernels
- Fourth interval: 40 kernels
- Residual heat: 10 kernels
Total kernels popped during cooking:(20 + 60 + 80 + 40 = 200)
Adding the residual heat kernels: (200 + 10 = 210)
Thus, the total number of pieces of popcorn Garrett has to eat is (\boxed{ 210 }).

| | |
|---|---|
| **Sens-Merging** | Okay, let's break down this problem step by step.
It's about popcorn popping over time,
and we need to figure out the total number of pieces Garrett has to eat.
First, let's list out the information given:
1. In the first 30 seconds, 20 kernels pop.
2. In the next 30 seconds, three times that amount pop.
So, 3 * 20 = 60 kernels.
3. In the following 30 seconds, the popping rate increases to four times the initial rate.
The initial rate was 20 per 30 seconds,
so four times that is 4 * 20 = 80 kernels.
4. In the final 30 seconds, the popping slows down to half the
rate of the previous 30 seconds. The previous rate was 80 per 30 seconds,
so half of that is 40 kernels.
5. After taking the pan off the heat, a quarter of
the number of kernels that popped in the final 30 seconds
also pop from the residual heat.
So, a quarter of 40 is 10 kernels.
Now, let's calculate the total number of popped kernels in each time interval:
1. First 30 seconds: 20 kernels
2. Next 30 seconds: 60 kernels
3. Following 30 seconds: 80 kernels
4. Final 30 seconds: 40 kernels
5. Residual heat: 10 kernels
To find the total number of kernels popped,
we add all these together:
$20 + 60 + 80 + 40 + 10 = 210$ kernels
So, Garrett has to eat 210 pieces of popcorn.
To determine the total number of pieces of popcorn Garrett has to eat,
we'll break down the problem step by step:
1. **First 30 seconds:** 20 kernels pop.
2. **Next 30 seconds:**
3 times the initial rate, so ($3 \times 20 = 6$) kernels pop.
3. **Following 30 seconds:** 4 times the initial rate, so ($4 \times 20 = 80$) kernels pop.
4. **Final 30 seconds:** Half the rate of the previous 30 seconds,
so ($\frac{80}{2} = 40$ ) kernels pop.
5. **Residual heat:** A quarter of the final 30 seconds' popping rate,
so( $\frac{40}{4} = 10$) kernels pop.
Now, let's add up all the popped kernels:
$[20 + 60 + 80 + 40 + 10 = 210]$
Therefore, the total number of pieces of popcorn Garrett has to eat is:[\boxed 210 ] |

**AIM**

To solve this problem, we need to calculate the number of popcorn kernels
that pop in each 30-second interval and then account for the additional
popping from the residual heat.
Let's break down the problem step-by-step:
1. In the first 30 seconds, 20 kernels pop.
2. In the next 30 seconds, 3 times the initial popping rate, so (3 times 20 = 60) kernels pop.
3. In the next 30 seconds,
4 times the initial popping rate, so (4 times 20 = 80) kernels pop.
4. In the final 30 seconds, half the rate of the previous 30 seconds,
so (\frac{80}{2} = 40) kernels pop.
After the pan is taken off the heat,
a quarter of the number of kernels that popped in the final 30 seconds
also pop from the residual heat.
So, (\frac{40}{4} = 10) additional kernels pop from the residual heat.
The total number of kernels that pop is the sum
of the kernels that pop in each interval plus the additional
kernels that pop from the residual heat.
Let's calculate this using Python code.

Python
```
# Initial popping rate
initial_rate = 20

# Popping rates in each 30-second interval
first_interval = initial_rate
second_interval = 3 * initial_rate
third_interval = 4 * initial_rate
fourth_interval = (4 * initial_rate) // 2

# Additional popping from residual heat
residual_popping = fourth_interval // 4

# Total number of kernels that pop
total_popped = first_interval + second_interval + third_interval
 + fourth_interval + residual_popping
print(total_popped)
```
The total number of pieces of popcorn that Garrett has to eat is (\boxed 200 ).

| | |
|---|---|
| **ACM** | Okay, let's break down this problem step by step.
It's about popcorn popping over time,
and we need to figure out the total number of pieces Garrett has to eat.
First, let's list out the information given:
1. In the first 30 seconds, 20 kernels pop.
2. In the next 30 seconds, three times that amount pop. So, 3 * 20 = 60 kernels.
3. In the following 30 seconds, four times the initial popping rate.
The initial rate was 20 per 30 seconds, so 4 * 20 = 80 kernels.
4. In the final 30 seconds, the popping rate is half of the previous 30 seconds.
The previous rate was 80 per 30 seconds, so half of that is 40 kernels.
5. After taking the pan off the heat,
a quarter of the number of kernels that popped in the final 30 seconds also
pop from the residual heat.
So, a quarter of 40 is 10 more kernels.
Now, let's add up all the popped kernels:
- First 30 seconds: 20
- Next 30 seconds: 60
- Following 30 seconds: 80
- Final 30 seconds: 40
- Residual heat: 10
Total popped kernels = 20 + 60 + 80 + 40 + 10 = 210
So, the total number of pieces of popcorn Garrett has to eat is 210.
To determine the total number of pieces of popcorn Garrett has to eat,
we'll break down the problem step by step:
1. **First 30 seconds:** 20 kernels pop.
2. **Next 30 seconds:** 3 times the initial rate,
so (3 times 20 = 60) kernels pop.
3. **Following 30 seconds:** 4 times the initial rate, so (4 times 20 = 80) kernels pop.
4. **Final 30 seconds:** Half the rate of the previous 30 seconds,
so ($\frac{80}{2} = 40$ ) kernels pop.
5. **Residual heat:** A quarter of the final 30 seconds' pops,
so ($\frac{40}{4} = 10$) additional kernels pop.
Adding all these together:
$[20 + 60 + 80 + 40 + 10 = 210]$
Therefore, the total number of pieces of popcorn is (\boxed 210 ). |

Table 13 presents the specific cases of ACM and other merging approaches on GSM8K dataset, showcasing the reasoning efficiency of our method. The comsumed tokens for each methods are 151 (Qwen, wrong answer), 737 (R1, correct answer), 356 (Sens-Merging, correct answer), 236 (AIM, wrong answer), **325** (ACM, correct answer) accordingly.

### A.6 Limitations

We acknowledge the potential limitations of our work. First, due to constrained computational resources, we did not conduct evaluations on extremely large-scale models, such as LLaMA-3.1-70B. Furthermore, all the models we tested are dense models, leaving evaluations on Mixture-of-Experts (MoE) models unexplored. From the perspective of model merging, our work focuses exclusively on merging models within the same architecture, without addressing heterogeneous model merging.

### A.7 Broader Impacts

**Positive Societal Impacts** By improving the efficiency and performance of LLMs in complex reasoning tasks while controlling output length, this research has the potential for significant positive societal impacts. More efficient and capable reasoning LLMs can accelerate scientific discovery, improve complex problem-solving in various domains (e.g., medical diagnosis, engineering design), and enhance educational tools by providing more nuanced and detailed explanations. The ability to generate concise yet accurate summaries of long reasoning chains through the L2S framework, as improved by ACM, could make advanced LLM capabilities more accessible and less computationally expensive, broadening their application in resource-constrained environments. This could lead to wider adoption of sophisticated AI assistants, potentially increasing productivity and innovation across industries and making advanced AI tools available to a broader range of users.

**Negative Societal Impacts**    Despite the potential benefits, the advancements presented also carry potential negative societal impacts. Improved reasoning and efficiency in LLMs, particularly the ability to condense complex information, could be leveraged for malicious purposes, such as generating highly convincing disinformation, crafting sophisticated phishing attacks, or automating the creation of deceptive content at scale. The enhanced ability to process and condense information might also raise privacy concerns if applied to sensitive data without robust safeguards. Furthermore, while the paper focuses on technical merging, the resulting models' deployment could exacerbate existing societal biases present in the training data or the models being merged, potentially leading to unfair or discriminatory outcomes in decision-making processes where these models are applied. The increased efficiency might also contribute to the concentration of power in the hands of those with access to and control over such advanced AI technologies. Mitigation strategies, such as promoting responsible development and deployment guidelines, developing methods for detecting malicious AI outputs, and addressing fairness and bias, are crucial to address these risks.

