# OpenReview forum: "Activation-Guided Consensus Merging for Large Language Models"
_NeurIPS.cc/2025/Conference — NeurIPS 2025 poster_

### Official Review · Reviewer_W7BU · 2025-06-21

**Clarity:** 3
**Significance:** 3
**Originality:** 3
**Rating:** 5
**Confidence:** 4

**Summary:**

The paper proposes Activation-Guided Consensus Merging (ACM), a novel method for merging Large Language Models (LLMs) by dynamically weighting layer contributions based on mutual information (MI) between activations of pre-trained and fine-tuned models. Unlike traditional merging techniques that assume uniform layer importance, ACM identifies task-specific critical layers, reducing redundancy while preserving performance. Experiments on Long-to-Short (L2S) reasoning and general merging tasks demonstrate that ACM improves reasoning accuracy (e.g., +1.3 points on Qwen-7B) while significantly shortening responses (e.g., -55.3% length). The method is plug-and-play, requiring no additional training or gradients, and outperforms baselines like TIES-Merging and DARE.

**Questions:**

1. Why use mutual information instead of other similarity metrics? The paper argues MI captures redundancy, but alternatives (e.g., cosine similarity, KL divergence) might offer different trade-offs in efficiency or interpretability.

2. How does ACM handle conflicting tasks in multi-task merging? The experiments focus on L2S and single-task merging. Can ACM resolve conflicts when merging models fine-tuned for opposing objectives (e.g., conciseness vs. verbosity)?

3. Is the improvement statistically significant? Accuracy gains (e.g., +1.3 points) are modest. Were significance tests conducted to rule out random variation?

**Ethical Concerns:**

["NO or VERY MINOR ethics concerns only"]

**Final Justification:**

This is an interesting work that introduces a novel technique. Given its value, I would like to see it made visible to the broader community.

**Limitations:**

While the hyperparameter t is claimed to be robust, its optimal value may vary across tasks (e.g., coding vs. math reasoning). No adaptive tuning strategy is proposed.

Results for 32B models show diminishing returns in length reduction. The paper does not address whether ACM’s benefits hold for trillion-parameter models or on-device LLMs.

**Paper Formatting Concerns:**

No.

**Quality:**

3

**Strengths And Weaknesses:**

1. Evaluated primarily on Qwen and LLaMA families; unclear how ACM performs on architectures with significantly different layer structures (e.g., MoE models).

2. Relies on a curated dataset (s1K) for activation alignment. Performance may degrade with poorly representative or biased calibration data.

3. How does the system 1/2 discussion related to this work "Visual agents as fast and slow thinkers" ICLR? Both work has similar motivation.

---

> ### Author Rebuttal · Authors · 2025-07-30
>
> We sincerely appreciate the time and effort you put into reviewing our paper. We appreciate the insightful feedback and address your concerns as follows:
>
> ---
>
> **R4W1:** MoE architecture
>
> **R4A1:** We would kindly clarify that as we stated in the end of section 2, Mixture-of-Experts (MoE) merging strategies require fundamental architectural modifications, including the incorporation of expert routing mechanisms and gating networks, which extend beyond the scope of parameter-space merging techniques addressed in this work [1]. While MoE represents a related research direction, its dynamic expert selection mechanism and architectural adaptations fundamentally distinguish it from the static parameter merging approaches investigated herein. We would like to note that the baseline methods compared in this paper didn‘t consider MoE merging either.
>
> [1] Branch-Train-MiX: Mixing Expert LLMs into a Mixture-of-Experts LLM. COLM 2024
>
> **R4W2:** Other calibration dataset
>
> **R4A2:** To alleviate the reviewers' concerns, we have also chosen the recently released LIMO[2] dataset as the calibration dataset for our experiments. LIMO challenges conventional wisdom in mathematical reasoning by showing that models can achieve superior performance with significantly less, yet higher quality, training data. In line with the settings applied in s1K, we conducted experiments using LIMO on GSM8K and Minerva Math, and the results are presented below:
>
> |                | GSM8K: Accuracy (Length) | Minerva Math: Accuracy (Length) |
> | ------------------ | -------------------------- | --------------------------------- |
> | Qwen-Math-1.5B   | 75.9 (118.1)             | 11.4 (1036.8)                   |
> | DeepSeek-R1-1.5B | 76.6 (2743.3)            | 15.1 (6374.2)                   |
> | Task Arithmetic  | 74.5 (549.7)             | 21.0 (1671.0)                   |
> | ACM-TA-s1K       | 76.8 (438.0)             | 25.3 (1214.7)                   |
> | ACM-TA-LIMO      | 76.9 (604.9)             | 22.3 (1549.7)                   |
>
> Our method demonstrates good adaptability to the new calibration dataset and exhibits robustness.
>
> [2] LIMO: Less is More for Reasoning. COLM 2025
>
> **R4W3:** Comparison of system 1/2 discussion to previous work
>
> **R4A3:** System 2 architectures are fine-tuned with extended thinking chains to promote deliberate analysis through iterative self-assessment, error mitigation, and verification, albeit facing challenges related to redundancy. In our work, the concept of System 2 follows [3,4,5], which is consistent with the notion that "System 2 delineates a cognitive mode distinguished by deliberate, analytical, and consciously reasoned processes" mentioned in "Visual agents as fast and slow thinkers." Furthermore, these works also align in their definition of System 1. We appreciate the reviewer's reminder, providing us with a visual perspective on system 1/2. We will cite this paper in our related works section in the final version.
>
> [3] OpenAI. Openai o3-mini: Pushing the frontier of cost-effective reasoning. Arxiv 2025.01
>
> [4] Deepseek-r1: Incentivizing reasoning capability in llms via reinforcement learning. Arxiv 2025.01
>
> [5] From system 1 to system 2: A survey of reasoning large language models. Arxiv 2025.02
>
> **R4Q1&Q2-a:** Usage of Mutual Information
>
> **R4A4:** We present our rationale for utilizing mutual information from the following perspectives:
> **1) Definition and Application:** Mutual information quantifies the interdependence between two variables, reflecting the extent to which knowledge of one reduces uncertainty about the other. It is extensively employed in representation learning [6]. In the context of model merging, as outlined in Section 3.1, mutual information assesses the relationship between the weights of the FT and PT models. High mutual information indicates redundancy (e.g., both models excel at similar tasks), resulting in limited performance improvements and potential noise introduction. Conversely, low mutual information suggests orthogonal knowledge, facilitating the effective integration of complementary capabilities and enhancing generalization.
> **2) Evaluation of Alternative Metrics:** In our preliminary experiments, we considered KL divergence and cosine similarity to measure model relationships.
>
> * **KL divergence** is asymmetric; for instance, with P=[0.9,0.1] and Q=[0.5,0.5], we found $D_{KL}(P∥Q)≈0.368$ and $D_{KL}​(Q∥P)≈0.511$, leading us to exclude this metric.
> * **Cosine similarity**  exhibits substantial variability across layers, approximately ranging from 0.25 to 0.8, which can negatively impact performance, as prior studies emphasized that weight coefficient fluctuations should be minimal [7]. Furthermore, the range of cosine similarity is (-1, 1), complicating the interpretation of negative values.
>
> In summary, we selected mutual information as a robust tool for examining model relationships. Due to space limitations, we emphasized its relevance in model merging and our further experiments validate its effectiveness. We appreciate the reviewers' suggestions and will incorporate preliminary observations in the final version.
>
> [6] A Mutual Information Maximization Perspective of Language Representation Learning. ICLR 2020
>
> [7] Sens-Merging: Sensitivity-Guided Parameter Balancing for Merging Large Language Models. ACL 2024
>
> **R4Q2-b:** Illustration of ACM on “conciseness vs. verbosity”
>
> **R4A5:** In the design of our ACM framework, when merging models with substantially different response patterns—such as in L2S tasks—or across heterogeneous tasks like code generation and mathematical reasoning, we assign relatively lower merging weights when the mutual information between corresponding layers is high, and higher weights when the mutual information is low. This adaptive weighting scheme effectively emphasizes task-specific (non-shared) features while preserving shared knowledge, thereby enhancing the model's ability to generalize across diverse task domains.
> We would kindly clarify that, as we mentioned in section 4.1 “Main Results”，we observed that response length positively correlates with question difficulty. Furthermore, as shown in the Appendix A.3, the merged model retains reflective capabilities; however, reflection frequency has decreased due to the PT fast-thinking model. According to our careful case study, while the merged model maintains favorable reasoning ability, it avoids redundant reflection on simpler mathematical problems, such as those in GSM8K, thereby reducing response length.
>
> **R4Q3:** Analysis and Explanations of Gains on ACM
>
> **R4A6:** In the L2S task, for instance, we would kindly clarify that ACM improves on TIES and R1 by **4.2%** and **2.3%**, respectively. Our manuscript shows ACM significantly outperforms AIM on the 1.5B model, with a supplementary 14B AIM experiment yielding 75.5% accuracy compared to ACM's 77.1% (a **1.6%** increase). We will incorporate the  AIM results into Appendix A.2..
>
> |   | GSM8K | MATH500 | AIME | Avg. |
> | ----- | ------- | --------- | ------ | ------ |
> | AIM | 46,7  | 88.2    | 46.7 | 75.5 |
> | ACM | 92.6  | 88.6    | 50.0 | 77.1 |
>
> Meanwhile, we would like to remind that as clarified in section 2, L2S task prioritizes length reduction while maintaining accuracy [8,9]; ACM achieves a **73.5%** length reduction relative to R1 on the 1.5B model with minimal resources.
>
> It can be observed that our model merging approach ACM is more efficient, with higher performance and lower length.
>
> We report the average scores across **five runs** with different random seeds. In Section 5, a comparison is made between our method and reinforcement learning (RL) approaches. Despite the considerably higher training costs associated with RL, our method attains a comparable reduction in length.
>
> [8] O1-Pruner: Length-Harmonizing Fine-Tuning for O1-Like Reasoning Pruning.  Arxiv 2025/01
>
> [9] L1: Controlling How Long A Reasoning Model Thinks With Reinforcement Learning. Arxiv 2025/03
>
> **R4Q4:** Hyperparameter and model size limitations
>
> **R4A7:** In comparison to alternative approaches, our method requires searching only **one** hyperparameter, whereas other baseline methods typically necessitate multiple hyperparameter searches. Due to resource constraints, we were unable to conduct experiments on 100B-scale models; however, we have successfully performed experiments on 32B (mentioned section 4.1 and Appendix Table 5) and 1.5B (Page 7, Table 2) models as reported.
>
> We would like to reiterate our sincere appreciation for the reviewer’s helpful and thoughtful comments.

---

> > ### Comment · Reviewer_W7BU · 2025-08-01
> >
> > Thanks for the detailed responses. All my concerns are addressed. I would like to increase my rating.

---

### Official Review · Reviewer_Dbaq · 2025-06-27

**Clarity:** 3
**Significance:** 3
**Originality:** 3
**Rating:** 5
**Confidence:** 3

**Summary:**

This paper proposes Activation-Guided Consensus Merging (ACM), a plug-and-play method for merging large language models (LLMs) using layer-wise weighting based on mutual information (MI) between activations. Motivated by the observation that different layers contribute unequally to downstream task performance, ACM computes per-layer MI between the pre-trained and fine-tuned models and uses this to derive adaptive coefficients for merging. Unlike previous methods that require gradient computation or rely solely on activations from a single model, ACM introduces a symmetric MI-based strategy that balances parameter reuse and task-specific adaptation. Extensive experiments on Long-to-Short (L2S) reasoning and general merging tasks across various model scales (1.5B to 32B) demonstrate that ACM consistently improves accuracy while reducing redundancy, outperforming existing merging baselines.

**Questions:**

I list my concerns and questions in the "weaknesses" section.

Aside from those, I have one additional point of interest:
- Given that the method computes MI over shared inputs, how well does it handle merging models trained on different tasks or exhibiting divergent behaviors? It would be helpful to discuss whether MI remains a meaningful signal in such heterogeneous scenarios.

**Ethical Concerns:**

["NO or VERY MINOR ethics concerns only"]

**Final Justification:**

Thanks for the detailed responses. All my concerns are addressed.

**Limitations:**

yes

**Paper Formatting Concerns:**

None.

**Quality:**

3

**Strengths And Weaknesses:**

Strengths:

- The paper presents a well-motivated solution to the limitations of uniform merging strategies, introducing a mutual information (MI)-based method for assigning layer-wise coefficients in a principled way.

- The proposed approach is training-free and computationally efficient, which is particularly appealing for real-world settings where retraining is impractical.

- The experimental setup is comprehensive, covering a range of model sizes, architectures (e.g., Qwen, LLaMA), and task types including math, commonsense reasoning, and code generation. The method demonstrates consistent gains across these settings.

- The use of MI as a signal for balancing redundancy and task specificity is conceptually clear and supported by both theoretical rationale and empirical evidence.

- The method is compatible with existing merging frameworks such as TA and TIES, and appears to further enhance their performance, particularly in managing accuracy and response length trade-offs.

Weaknesses:

- The computation of MI-based merging coefficients relies on a calibration dataset and its clustering, which may introduce some sensitivity to the choice of data and its representativeness. A more detailed discussion of this point would be helpful.

- While the method is designed to be plug-and-play, the requirement to extract activations from both the pre-trained and fine-tuned models introduces added complexity compared to simpler, static baselines.

- The evaluation primarily focuses on long-to-short reasoning and mathematical tasks. Additional results on open-ended generation or instruction-following could help better understand the method’s generalization across a wider range of use cases.

---

> ### Author Rebuttal · Authors · 2025-07-30
>
> We sincerely appreciate the valuable advice from the reviewer. And we would like to address your concerns as follows:
>
> ---
>
> **R3W1:** Calibration dataset
>
> **R3A1:** In Section 4 “Models and Datasets”, we mentioned that we cluster the data in s1K and uniformly sample 10% (1000 \* 10% = 100 pieces) of the data for multiple experiments. This approach is intended to mitigate potential biases arising from uneven sampling. Additionally, in earlier experiments, we sampled varying sizes of calibration datasets for analysis, and the test results for GSM8K on the 7B model are presented below:
>
> | Data Pieces | 20     | 50     | 100   | 200    | 300   |
> | ------------- | -------- | -------- | ------- | -------- | ------- |
> | Accuracy  | 91.4   | 90.8  | 92.2 | 91.8  | 92.1  |
> | Length      | 623.63 | 624.17 | 538.3 | 585.65 | 603.0 |
>
> Our observations indicate that once the number of calibration data points reaches 100, accuracy remains high while length remains short. We also analyzed the weight coefficients at this threshold. When the number of data is below 100, the distribution of weight coefficients fluctuates significantly, with the top five layers showing considerable variation. Conversely, when the count exceeds 100, these characteristics stabilize. This trend is similar across models such as 1.5B and 14B. Thus, we advocate for a 10% sampling rate in our experimental design. For larger models (e.g., those exceeding 100B), additional calibration data may be required; however, resource limitations prevented us from further experimentation. We posit that activation-based methods necessitate fewer data and computational resources compared to supervised fine-tuning or reinforcement learning.
>
> We also present the results of the 7B model using different sampling seeds as follows:
>
> |         | seed a | seed b | seed c |
> | ---------- | -------- | -------- | -------- |
> | Accuracy | 91.9 | 91.7  | 92.4   |
> | Length   | 558.33 | 553.58 | 541.41 |
>
> It is evident that our clustering and sampling methods are both reasonable and robust.
>
> Additionally, we investigated the effects of varying the number of clusters. Our results demonstrate that random clustering results in suboptimal performance, indicating possible imbalance in the dataset's distribution. Notably, with and beyond 20 clusters, we observed improved accuracy and reduced length, suggesting that clustering then sampling effectively mitigates data imbalance.
>
> | Clusters | random | 10    | 20    | 30    |
> | ---------- | -------- | ------- | ------- | ------- |
> | Accuracy | 88.6   | 91.1 | 91.8  | 91.6 |
> | Length   | 643.4  | 602.2 | 538.3 | 586.7 |
>
> We sincerely appreciate the reviewers' suggestions and will incorporate these experiments into the appendix.
>
> **R3W2:** Computational overhead as a plug-and-play approach
>
> **R3A2:**  According to our analysis, the ACM method demonstrates exceptional computational efficiency on the experimental equipment mentioned in the paper: the 1.5B model experiment requires only about **40 seconds** on **CPU**, while the 7B and 14B models need only about **1-1.5 minutes**. Considering the significant accuracy improvement brought by ACM (such as **2.3%** improvement over R1 for the 1.5B model) and the substantial length reduction (**73.5%** reduction compared to R1), the required computational overhead is negligible. Furthermore, some baseline methods, such as Sens-Merging and AIM, also require additional calibration calculations. As analyzed in section 5, Figure 5, compared to resource-intensive training methods, ACM can easily achieve significant length reduction effects at extremely low cost.
>
> We further supplement the results with fine-tuning of the DeepSeek-R1-1.5B model (optimized for slow thinking) on s1K's COT data (optimized for fast thinking), which took approximately **10 minutes** for three epochs on a **GPU**. The experimental results are presented below:
>
>  |                | MATH: Accuracy(Length) | GSM8K: Accuracy(Length)| Time Consumption |
> | ------------------- | ------------------------ | ------------------------- | ----------------- |
> | Qwen2.5-Math-1.5B | 36.2 (411.0)          | 75.9 (118.1)           | NA               |
> | DeepSeek-R1-1.5B  | 69.6 (4508.2)         | 76.6 (2743.3)          |       NA         |
> | Qwen2.5-SFT       | 70.6 (4372.5)         | 77.5 (2536.3)          | **10 minutes**     |
> | ACM(ours)       | 71.4 (1235.6)         | 78.4 (962.1)           | **40 seconds**      |
>
> It can be observed that our model merging approach ACM is more efficient, with higher performance and lower length.
>
> **R3W3:** Extension Experiments
>
> **R3A3:** For a fair comparison, we followed the task selection of the baseline methods and conducted experiments on the L2S tasks (including both simple and complex reasoning tasks, code generation tasks), as well as general model merging tasks (math and code model merging tasks). We appreciate the reviewer's suggestion; due to limitations in API resources, we selected the IFEval dataset [2]—a benchmark for instruction following—to evaluate our method. The experimental results are as follows:
>
> |        | prompt-level   | instruction-level |
> | ---------- | ---------------- | ------------------- |
> | TIES     | 18.48          | 30.58             |
> | ACM-TIES | 20.15 (+1.67) | 31.06 (+0.48)    |
>
> We find that on the instruction-following dataset, ACM can further enhance the model merging performance of TIES, thereby further validating the effectiveness of our ACM approach.
>
> **R3Q1:** Mutual information on heterogeneous scenarios
>
> **R3A4:** Mutual information quantifies the interdependence between two random variables, reflecting the extent to which knowledge of one reduces uncertainty about the other. It is extensively employed in representation learning [1]. In the context of model merging, as outlined in Section 3.1, mutual information assesses the relationship between the weights of the FT and PT models. High mutual information indicates redundancy (e.g., both models excel at similar tasks), resulting in limited performance improvements and potential noise introduction. Conversely, low mutual information suggests orthogonal knowledge (e.g., code and math tasks), facilitating the effective integration of complementary capabilities and enhancing generalization.
>
> Therefore, in the design of our ACM framework, when merging models with substantially different response patterns—such as in L2S tasks—or across heterogeneous tasks like code generation and mathematical reasoning, we assign relatively lower merging weights when the mutual information between corresponding layers is high, and higher weights when the mutual information is low. This adaptive weighting scheme effectively emphasizes task-specific (non-shared) features while preserving shared knowledge, thereby enhancing the model's ability to generalize across diverse task domains.
>
> We sincerely thank the reviewer for the insightful and constructive feedback. We hope our responses could adequately addressed the reviewer's concerns.
>
> References:
>
> [1] A Mutual Information Maximization Perspective of Language Representation Learning. ICLR 2020
>
> [2] Instruction-Following Evaluation for Large Language Models.  Arxiv 2023.11

---

> > ### Author Response · Authors · 2025-08-05
> > **Official Comment by Authors**
> >
> > Dear Reviewer Dbaq,
> >
> > We hope this message finds you well.
> >
> > As the rebuttal period deadline is approaching, we would greatly appreciate it if you could kindly acknowledge and respond to our rebuttal for paper NeurIPS 9813. We are looking forward to receiving any further questions or suggestions for improvement. If your concerns have been resolved, could you raise your score to support our work?
> >
> > We sincerely thank you for your support.
> >
> > Kindest regards, Authors of NeurIPS 9813

---

> > ### Comment · Reviewer_Dbaq · 2025-08-05
> >
> > Thanks for the detailed responses. All my concerns are addressed, and I believe the paper now provides more clarity and stronger contributions. I would like to increase my rating.

---

### Official Review · Reviewer_fHAX · 2025-06-28

**Clarity:** 3
**Significance:** 2
**Originality:** 2
**Rating:** 3
**Confidence:** 4

**Summary:**

The paper studies model merging over LLMs, with particular focus on the Long-to-Short task aiming to reduce the “reasoning trace” of reasoning models without waiving accuracy. To do so, they leverage task vectors, differences between finetuned models and their pretrained counterpart. The novelty of the paper stems in finding layer-specific merging coefficients that are a function of the mutual information between the activations of the task-specific model and the pretrained one.  The approach is extensively evaluated on long-to-short over Qwen2.5, Qwen2.5-Math and corresponding DeepSeek-R1 versions over GSM8K, MATH500, Minerva Math, Olympiadbench, College Math and AIME 2024. While LLama2-7B, MammoMath and CodeLlama-7b-hf over GSM8K, HellaSwag, MBPP-Pro are considered for general merging.

**Questions:**

- L100 says “arithmetic merging operates fine-tuned features as directional task vectors in parameter space”. What does this mean? I suggest rephrasing
- Is there a way to assess to what degree does the result depend on the quality/size of the calibration dataset?
- How significant are the gains? being somewhat marginal, are these expected to be consistent across different seeds and setups?

Overall, given the added requirements with respect to standard baselines, the method needs to yield convincingly better results while ensuring the significance of these ones.

**Ethical Concerns:**

["NO or VERY MINOR ethics concerns only"]

**Final Justification:**

The paper presents a novel model merging approach mainly aiming to reduce the length of the models' output while preserving their accuracy. I raised the following concerns: minimal gains with respect to existing data-free baselines, incrementality of the approach and unconvincing motivations, along with some typos and inconsistencies. The rebuttal partially addressed some of these concerns, but I still feel the first and most important one to hold true: while ACM shows ~4% improvement over TIES in some cases, gains are marginal for LLaMA and TIES remains highly competitive for the stated goal of length reduction. The need for task-specific calibration data limits applicability, and the main stated reason of qualitative difference between the proposed method and prior work (not necessitating complex gradient computation) offers only modest practical benefit given merging’s one-off cost. Overall, the improvements do not, in my view, justify the added complexity and requirements. I maintain my borderline reject score.

**Limitations:**

yes

**Quality:**

2

**Strengths And Weaknesses:**

### Strengths

- The paper is fairly clearly written, with clear methodology and detailed experimental setup. The code  is also provided for reproducibility and expected to be released upon acceptance.
- The empirical evaluation is extensive, covering a diverse set of both reasoning and standard tasks for several different open state-of-the-art LLMs. The set of baselines is comprehensive enough.
- The paper tackles an interesting and felt problem, as reducing the reasoning trace without waiving accuracy has immediate impactful applications, saving both computation and time.

### Weaknesses

- The gains seem marginal. When compared with AIM, the method obtains the same length reduction and an increase of 0.8% in accuracy. What is most striking, however, is that ACM-TIES with respect to standard TIES only results in +1% accuracy and +2.3% length on QWEN-1.5B and +0.08% accuracy on Lllama, despite requiring calibration data and added complexity.
- The proposed approach seems to be incremental with respect to Sens-Merging [1], with the main difference stemming from a different way to compute the sensitivity of the parameters, as the proposed approach uses mutual information and Sens-merging using gradients. The paper only motivates the benefit of the approach with respect to Sens-merging as not requiring “complex gradient calculation”; I am not sure what complex here might mean. is it computationally expensive? error-prone?
- Unconvincing motivation. I did not fully grasp why the mutual information between the activations of the pretrained model and those of the task-specific models is a meaningful quantity to be considered. Some additional explaination or theoretical/quantitative analysis might help understanding why this is a good idea in the first place.
- Some minor inconsistencies.
    - the paper claims: “In the specific case sigma is the activation function, the gradient is {0, 1}”; this is only true for some activation functions, e.g. ReLU, and does not hold in general.
    - There are often inconsistenc spaces before/after equatons. I suggest removing all empty lines surrounding equation blocks in the source code.

[1] Liu, Shuqi, et al. "Sens-Merging: Sensitivity-Guided Parameter Balancing for Merging Large Language Models." *arXiv preprint arXiv:2502.12420* (2025).

---

> ### Author Rebuttal · Authors · 2025-07-30
>
> We sincerely thank the reviewer for the constructive suggestions. We would like to address your concerns as follows:
>
> ---
>
> **R2W1&Q3-a&Q4:** Analysis and Explanations of Gains on ACM
>
> **R2A1:** In the L2S task, we would kindly clarify that ACM improves on TIES and R1  by **4.2%** and **2.3%** respectively, not 1% as the reviewer stated. Our Table 2 in the manuscript shows ACM significantly outperforms AIM on the 1.5B model, with a supplementary 14B AIM experiment yielding
> 75.5% accuracy compared to ACM's 77.1% (a 1.6% increase). We will incorporate the  AIM results into Appendix A.2.
>
> |    | GSM8K | MATH500 | AIME | Avg. |
> | ----- | ------- | --------- | ------ | ------ |
> | AIM | 46,7  | 88.2    | 46.7 | 75.5 |
> | ACM | 92.6  | 88.6    | 50.0 | 77.1 |
>
> Meanwhile, we would like to remind that as clarified in section 2, *L2S task prioritizes length reduction while maintaining accuracy[2,3]*. It is expected that responses are slightly longer on the L2S task due to the integration of long CoT model, compared to short CoT model. ACM achieves a **73.5%** length reduction relative to R1 on the 1.5B model with minimal resources.
>
> As for resource consumption, as outlined in Section 4.1 Table 1, we employed merely 10% of the calibration data. In our experiments, merging a 1.5B model on a **CPU** required approximately **40 seconds**, whereas fine-tuning the R1-1.5B model (optimized for slow thinking) with COT data (optimized for fast thinking)  from S1K took around  **10 minutes** for three epochs on **GPU**.
>
> |                  | MATH  Accuracy(Length) | GSM8K Accuracy(Length) |
> | ------------------- | ------------------------ | ------------------------- |
> | Qwen2.5-Math-1.5B | 36.2 (411.0)          | 75.9 (118.1)           |
> | DeepSeek-R1-1.5B  | 69.6 (4508.2)         | 76.6 (2743.3)          |
> | Qwen2.5-SFT       | 70.6 (4372.5)         | 77.5 (2536.3)          |
> | ACM     | 71.4 (1235.6)         | 78.4 (962.1)           |
>
> The model merging approach proves to be more efficient, delivering higher performance with reduced length. In Section 4, we highlighted the DPO performance on the 7B model, demonstrating that model merging is both resource-efficient and effective. Section 5 compares our method to popular RL L2S approaches. Despite the higher training costs of RL, our method achieves a similar reduction in length. We report average scores across **five** runs with different random seeds.
>
> In general tasks, smaller gains align with UniTE[1]'s recommendation that model accuracy differences should be less than 15%, as the performance disparity among models in the pool is substantial, resulting in limited improvements.
>
> [1] Determine-Then-Ensemble: Necessity of Top-k Union for Large Language Model Ensembling. ICLR 2025
>
> [2] O1-Pruner: Length-Harmonizing Fine-Tuning for O1-Like Reasoning Pruning.  Arxiv 2025/01
>
> [3] L1: Controlling How Long A Reasoning Model Thinks With Reinforcement Learning. Arxiv 2025/03
>
> **R2W2:** Difference compared to Sens-Merging
>
> **R2A2:**  ACM and Sens-Merging are fundamentally distinct in methodology and design. **1)**  ACM leverages activation information—commonly used in model quantization [4]—and requires only forward propagation, whereas Sens-Merging depends on gradient computation, necessitating both forward and backward passes. This results in higher computational overhead for Sens-Merging. **2)**  ACM assesses model information correlation through mutual information, whereas Sens-Merging requires multiple task-specific and cross-task scaling hyperparameters, adding complexity and sensitivity. In contrast, ACM depends on a single hyperparameter, demonstrating robustness and simplicity. **3)** Experimental results demonstrate that ACM consistently outperforms Sens-Merging, validating its effectiveness.
> Due to space constraints, we focused on ACM principles in the manuscript but will incorporate more details on baseline methods like Sens-Merging in the final version.
>
> [4] AWQ: Activation-aware Weight Quantization for LLM Compression and Acceleration. MLSys 2024
>
> **R2W3:** Usage of Mutual Information
>
> **R2A3:** We present our rationale for utilizing mutual information from the following perspectives:
>
> **1) Definition and Application:** Mutual information(MI) quantifies the interdependence between two variables, reflecting the extent to which knowledge of one reduces uncertainty about the other. It is extensively employed in representation learning [5]. In model merging, as outlined in Section 3.1, MI assesses the relationship between the weights of the FT and PT models. High MI indicates redundancy (e.g., both models excel at similar tasks or response styles), resulting in limited performance improvements and potential noise introduction. Conversely, low MI suggests orthogonal knowledge, facilitating the effective integration of complementary capabilities and enhancing generalization.
>
> **2) Evaluation of Alternative Metrics:** In our preliminary experiments, we considered KL divergence and cosine similarity to measure model relationships.
>
> * **KL divergence** is asymmetric; for instance, with P=[0.9,0.1] and Q=[0.5,0.5], we found $D_{KL}(P∥Q)≈0.368$ and $D_{KL}​(Q∥P)≈0.511$, leading us to exclude this metric.
> * **Cosine similarity**  exhibits substantial variability across layers, approximately ranging from 0.25 to 0.8, which can negatively impact performance, as prior studies emphasized that weight coefficient fluctuations should be minimal [6]. Furthermore, the range of cosine similarity is (-1, 1), complicating the interpretation of negative values.
>
> We chose MI as a robust tool for examining model relationships. Due to space limitations, we emphasized its relevance in model merging and our further experiments validate its effectiveness. We appreciate the reviewers' advice and will incorporate preliminary observations in the final version.
>
> [5] A Mutual Information Maximization Perspective of Language Representation Learning. ICLR 2020
>
> [6] Sens-Merging: Sensitivity-Guided Parameter Balancing for Merging Large Language Models. ACL 2024
>
> **R2W4:** Activation Theoretical Analysis
>
> **R2A4:** We sincerely thank the reviewer for this insightful comment. Our intention was to use ReLU as an illustrative example to clearly present the core mechanism of our proposed method.  In our analysis, the central quantity of interest is $S(W) := sup_x sup_∇ ⟨∇, Δ W x⟩$, where the supremum is taken over the set of all possible gradients ∇ of the activation function. For this quantity to be well-defined and serve as a meaningful bound, the essential requirement is that the set of gradients must be **bounded.**
>
> * In the specific case of ReLU, the set of gradients is simply {0, 1}, making the characterization of $S(W)$ very straightforward. We used this case to build intuition.
> * For more general activation functions, such as GELU and SwiGELU, the gradient ∇ is not binary but belongs to a continuous, bounded interval (approximately [-0.17, 1.17]). Because this set is bounded, the sup_∇ operation is still well-defined, and our formulation of $S(W)$ remains valid and finite.
>
> Therefore, our core argument does not rely on the gradient being binary, but on the fact that for almost all modern activation functions, the gradient is bounded. This ensures that $S(W)$  is a meaningful quantity that can be analyzed.
>
> We appreciate the reviewer's feedback and will revise this section to introduce the quantity $S(W)$ and a bounded gradient. We will adjust space format as well.
>
> **R2Q1:** Explanation of L100
>
> **R2A5:** The task vector is defined as  $δ_i = θ_i − θ_0$, $θ_i$ usually refers to the weights of fine-tuned models, and $θ_0$ is that of the pretrained model. Task vectors highlight fine-tuned features. Arithmetic merging, including average or weighted averaging (TA), directly operates on these task vectors.
>
> **R2Q2&Q3-b:** Calibration dataset
>
> **R2A6:** In Section 4 “Models and Datasets”, we mentioned that we cluster the data in s1K and uniformly sample 10% (100 pieces) of the data for multiple experiments. This approach is intended to mitigate potential biases arising from uneven sampling. Additionally, in earlier experiments, we sampled varying sizes of data for analysis, and the test results for GSM8K on the 7B model are presented below:
>
> | Data Pieces | 20     | 50     | 100   | 200    | 300   |
> | ------------- | -------- | -------- | ------- | -------- | ------- |
> | Accuracy| 91.4   | 90.8  | 92.2 | 91.8  | 92.1  |
> | Length      | 623.63 | 624.17 | 538.3 | 585.65 | 603.0 |
>
> Our results show that accuracy stabilizes and response length remains short once calibration data reaches 100 samples. We analyzed the weight coefficients at this threshold. Below 100 data points, the distribution fluctuates significantly, particularly in the top five layers. However, when the count exceeds 100, these characteristics stabilize. This trend holds across 1.5B and 14B models, supporting our use of a 10% sampling rate. For larger models (e.g., >100B), more data may be needed, though resource constraints limited further study. We posit that activation-based methods necessitate fewer data and computational resources compared to SFT or RL.
>
> We also present the results of the 7B model using different seeds as follows:
>
> |         | seed a | seed b | seed c |
> | ---------- | -------- | -------- | -------- |
> | Accuracy | 91.9 | 91.7  | 92.4   |
> | Length   | 558.33 | 553.58 | 541.41 |
>
> It is evident that our clustering and sampling methods are both reasonable and robust. We sincerely appreciate the reviewers' suggestions and will incorporate these experiments into the appendix.
>
> Again, we sincerely appreciate the valuable suggestions provided by the reviewers. If our explanations can assist the reviewers in better understanding our paper, we would be delighted. Should the reviewers have any further questions, we welcome further discussion.

---

> > ### Comment · Reviewer_fHAX · 2025-08-04
> >
> > I thank the authors for their detailed rebuttal, addressing some of my concerns (rationale behind the metric, the relation between sigma and the gradient, calibration dataset size vs performance). However, I am still unconvinced in some aspects.
> >
> > In particular, the authors correctly point out one oversight from my side, as actually ACM improves TIES by ~4% according to the tables. However, the marginal gain for LLaMa (+0.08) is still valid. Moreover, if the main goal is to reduce the length, as the authors claim in the rebuttal, then TIES is still an extremely competitive baseline due to the close accuracy and roughly -75.5% token length (vs 73.5 from ACM).
> >
> > I am also still not fully convinced about this "complexity" deriving from the gradient computation in Sens-Merging. In what way does this computation pose an issue? Is it more costly? I feel like my question was not answered.

---

> ### Author Response · Authors · 2025-08-05
> **Response to the reviewer**
>
> Thanks for the reviewers' feedback. In response to the remaining questions, we offer the following clarifications:
>
> 1. The reviewers noted that TIES demonstrates better length reduction in the 1.5B L2S model. While we acknowledge this, it's important to assess method effectiveness by considering both accuracy and length across different model sizes, such as 7B and 14B. In the 7B model, ACM achieves higher accuracy than Ties, with length reductions of 55.3% and 53%, respectively. For the 1.5B model, ACM exhibits greater length reduction on challenging datasets like AIME24, Olympiad Bench, and College Math compared to TIES. **We emphasize that ACM effectively balances accuracy and length on the L2S task**.
>
>    ACM, as a low-cost method, can be effectively combined with approaches such as TA and TIES to enhance performance. If users of the 1.5B L2S prioritize both length reduction and performance, we recommend ACM. For those focused solely on maximum length compression, TIES may be a more suitable option. We hope the reviewers will assess ACM's effectiveness based on comprehensive experimental results.
>
> 2. Regarding the LLaMA experiments, we explained in our rebuttal that the significant accuracy difference between MammoMATH (41.55) and Codellama-7B (25.47) limits performance improvements after merging [1][2]. Besides, similar marginal improvements are observed in broader experiments like Sens-Merging and AIM-Merging, implying some common magnitude of model merging. We refer to the original texts of AIM [3] and Sens-Merging for further details.
>
> | Model (LLaMA series) | MBPP accuracy  |
> | ----------------------- | ---------------- |
> | chat                  | 27.60          |
> | math                  | 31.80          |
> | code                  | 31.60          |
> | ties                  | 29.20          |
> | AIM-Merging(TIES)     | 29.20（**+0.00**） |
>
> | Model (LLaMA series) | Avg. accuracy  |
> | ----------------------- | ---------------- |
> | chat                  | 31.18          |
> | math                  | 37.42          |
> | code                  | 31.60          |
> | DARE                  | 40.13          |
> | Sens-merging (DARE)   | 40.35（**+0.22**） |
>
> Although our model versions differ from those used in Sens-Merging and AIM-Merging, the phenomenon of limited improvement is commonly observed. Given the variability in the availability of open-source models, the performance disparity of the models we are working with is even greater, which underscores the effectiveness of the ACM method.
>
> 3. We explained that ACM only requires a forward pass, whereas Sens-Merging involves both forward and backward processes, making the computation for Sens-Merging more complex. In terms of time overhead, in our experiments and replication process, ACM takes approximately **40 seconds** on the 1.5B model, while Sens-Merging takes about **90 seconds.**
>
> [1] Breaking the Ceiling of the LLM Community by Treating Token Generation as a Classification for Ensembling. EMNLP 2024
>
> [2] Determine-Then-Ensemble: Necessity of Top-k Union for Large Language Model Ensembling. ICLR 2024
>
> [3] Activation-Informed Merging of Large Language Models. Arxiv 2025.

---

> > ### Comment · Reviewer_fHAX · 2025-08-05
> >
> > I thank the authors for their answer.
> >
> > I am not convinced about the difference in accuracy being reason enough to justify such marginal gains. In my understanding, the method barely outperforms general data-free merging techniques such as TIES while requiring task-specific data, and in some cases is actually less advisable for the very task for which it was initially proposed.
> > The fact that marginal improvements are common in the literature is, in my opinion, more an issue with the literature than an intrinsic property of the investigated problems. If a method outperforms simple baselines by a small margin on some set of benchmarks and architectures, it is very likely that the same method will not still rank preferably on different ones, and hardly make for a contribution in actually solving the problem.
> >
> > Regarding the motivation of the method with respect to Sens-Merging, I understand that the gradient computation results in 90s for merging instead of 40s. Given that merging is a one-off step that is performed once in place of costly finetuning or RLHF, this doesn't seem like a gamechanger. While I clearly appreciate faster and more efficient methods, the fact that the main differentiation between the proposed methods and prior work is based on the work not necessitating "complex backpropagation" makes the motivation feel somewhat artificial.
> >
> > > Nonetheless, these methods either overlook inter-model relationships by solely focusing on PT model’s activations or necessitate complex backpropagation.
> >
> > Overall, comparing the method with simple and requirement-free baselines, I find the benefits of the method not to justify the added complexity and requirements. I confirm my initial score.

---

### Official Review · Reviewer_7bwQ · 2025-07-03

**Clarity:** 3
**Significance:** 3
**Originality:** 3
**Rating:** 5
**Confidence:** 3

**Summary:**

The paper proposes Activation-Guided Consensus Merging (ACM) to improve the integration of multiple LLMs through model merging. Existing merging techniques often assume uniform importance across model layers, ignoring the diverse functional roles of different layers. ACM addresses this by using mutual information (MI) between layer-wise activations of pre-trained and fine-tuned models to assign layer-specific merging coefficients, giving more weight to divergent task-specific layers and less to redundant ones. This approach is plug-and-play, requiring no gradient computations or retraining.

**Questions:**

Do you assume that the fine-tuning data for all models is available?
* If so, why would one apply model merging instead of simply fine-tuning on all the data, given that fine-tuning is usually not considered costly?
* If not, can you provide an example of an application scenario where the datasets are not available, but the fine-tuned weights are?

**Ethical Concerns:**

["NO or VERY MINOR ethics concerns only"]

**Final Justification:**

The authors addressed my concerns in their rebuttal, and I have increased my score from 4 to 5.

**Limitations:**

I don't see obvious negative societal impact of this work.

**Paper Formatting Concerns:**

No Paper Formatting Concerns

**Quality:**

3

**Strengths And Weaknesses:**

**Strengths**
* The paper writing is overall clear and easy to follow.
* The proposed model merging technique is well-motivated and reasonably principled.
* The experimental performance is well acceptable, even compared with strong baselines such as Deepseek r1 and Qwen-2.5.

**Weakness**
I don't see significant weakness.

---

> ### Author Rebuttal · Authors · 2025-07-30
>
> We greatly appreciate the reviewer’s helpful feedback and recognition of our contributions. We would like to address your concerns as follows:
>
> ---
>
> **R1Q1:** Availability of fine-tuning data
>
> **R1A1:** We address the reviewers' considerations as follows:
>
> * **Data Acquisition Challenges:** For fine-tuned models, the associated fine-tuning data is often inaccessible due to concerns regarding data privacy, ownership, and compliance with regulatory policies. For instance, while the HuggingFace platform hosts a large number of open-weights models, such as Qwen series and DeepSeek R1 series, the corresponding fine-tuning datasets are frequently unavailable or undisclosed.
> * **Orthogonality of Model Merging and Fine-Tuning:** Model merging and fine-tuning are orthogonal processes. After fine-tuning, model merging can still be performed. For example, we can merge models that have been fine-tuned in specific domains, such as fast-thinking models (Qwen series) and slow-thinking models (R1-Distill series), thereby integrating their capabilities, obtain a model that possesses long-chain reasoning capability while maintaining concise overall responses.
> * **Comparative Time Efficiency:** Fine-tuning is comparatively more time-intensive than model merging. In our experiments, merging a 1.5B model on  **CPU** required approximately **40 seconds**, whereas fine-tuning the DeepSeek-R1-1.5B model (optimized for slow thinking) with COT data (optimized for fast thinking) from s1K took around **10 minutes** for **three epochs on GPU**. The experimental results are presented below:
>
>  |                | MATH: Accuracy(Length) | GSM8K: Accuracy(Length)| Time Consumption |
> | ------------------- | ------------------------ | ------------------------- | ----------------- |
> | Qwen2.5-Math-1.5B | 36.2 (411.0)          | 75.9 (118.1)           | NA               |
> | DeepSeek-R1-1.5B  | 69.6 (4508.2)         | 76.6 (2743.3)          |       NA         |
> | Qwen2.5-SFT       | 70.6 (4372.5)         | 77.5 (2536.3)          | 10 minutes      |
> | ACM(ours)       | 71.4 (1235.6)         | 78.4 (962.1)           | 40 seconds      |
>
>    It can be observed that the model merging approach is more efficient, with higher performance and shorter length. Additionally, we reported the performance of DPO on the 7B model in *Section 4, Table 1*, finding that model merging is a resource-efficient yet highly effective method.
>
> Once again, we greatly appreciate the reviewer’s valuable feedback and constructive suggestions.

---

> > ### Author Response · Authors · 2025-08-05
> > **Official Comment by Authors**
> >
> > Dear Reviewer 7bwQ,
> >
> > We hope this message finds you well.
> >
> > As the rebuttal period deadline is approaching, we would greatly appreciate it if you could kindly acknowledge and respond to our rebuttal for paper NeurIPS 9813. We are looking forward to receiving any further questions or suggestions for improvement. If your concerns have been resolved, could you raise your score to support our work?
> >
> > We sincerely thank you for your support.
> >
> > Kindest regards, Authors of NeurIPS 9813

---

> > > ### Author Response · Authors · 2025-08-07
> > > **Official Comment by Authors**
> > >
> > > Dear Reviewer 7bwQ,
> > >
> > > We hope this message finds you well.
> > >
> > > As the rebuttal period deadline is fast approaching (in less than 2 days), we would kindly appreciate it if you could acknowledge and respond to our rebuttal for paper NeurIPS 9813. We have responded to and solved your concerns. We are eager to address any further questions or suggestions you may have for improvement. If your concerns have been resolved, we would be grateful if you could consider raising your score or your confidence in support of our work.
> > >
> > > Thank you very much for your time and support.
> > >
> > > Kindest regards, Authors of NeurIPS 9813

---

> > ### Comment · Reviewer_7bwQ · 2025-08-08
> >
> > Thank you to the authors for your responses, which I found convincing. I have raised my score to 5.

---

### Note · Authors · 2025-08-13

We sincerely thank the AC for your dedicated efforts in managing the review process. We greatly appreciate the reviewers’ active engagement. After rebuttal, reviewers 7bwQ, Dbaq, and W7BU increased scores from 4 to 5 or higher, and fHAX improved sub-scores like "quality," reflecting strong recognition and encouragement. Below, we summarize key rebuttal responses and manuscript improvements.

**Key Contributions of Our Work**

1) We theoretically analyze the correlation between activation space and weight salience, propose a novel method ACM, which leverages MI to compute the weight coefficients of each layer.
2) L2S experiments demonstrate ACM's effectiveness in maintaining performance and reducing redundancy without any training cost. E.g., ACM achieves up to **55.8%** shorter sequences with a **+1.3** accuracy gain across six 7B-model benchmarks.
3) General merging tasks further validate ACM’s soundness and efficacy.

The results show that ACM serves as an effective plug-and-play solution in model merging field.

**Major Rebuttal Discussions & Resolutions**

1. **Calibration Data** (reviewer fHAX, Dbaq, W7BU): We add ablation studies on data volume, random seeds, and cluster numbers, and validate ACM’s robustness on the new LIMO dataset.
2. **Gains Analysis** (reviewer fHAX, W7BU): On L2S, ACM outperforms baselines in compression-accuracy trade-off across 1.5B–32B models. For general merging, we analyze modest gains via base model gaps. We also respectfully clarify that the paper included comparisons with RL-based methods, validating the ACM's effectiveness.
3. **Explanation of MI** (reviewer fHAX, Dbaq, W7BU): We justify the use of MI via theoretical and preliminary experimental analysis, and refine the theoretical derivation for clarity.
4. **Time Consumption** (reviewer 7bwQ, Dbaq): We include a comparison on 1.5B models: ACM takes **40s** vs. **10 mins** for SFT, with **30%** the length, indicating superior efficiency.

**Reviewer-Specific Responses & Improvements**

* fHAX: We explain the differences between ACM and baselines in terms of computational procedures and the amount of hyperparameters.
* Dbaq: We add instruction-following experiments (e.g. IFEval dataset) to further validate our generalizability.
* W7BU: We will include additional reference to better explain System 1/2.

We believe all major concerns have been addressed, and the revisions strengthen the paper. We sincerely thank the AC and reviewers for the valuable feedback and guidance.

---

### Decision · Program_Chairs · 2025-09-17

**Decision:**

Accept (poster)

**Comment:**

This paper proposes a plug-and-play merging framework that determines layer-specific merging coefficients based on mutual information between activations of pre-trained and fine-tuned models. The proposed method, called activation-guided consensus merging (ACM), preserves task-specific capabilities without requiring gradient computations or additional training. The experimental evaluation on Long-to-Short (L2S) and general merging tasks demonstrates that ACM outperforms baseline methods.

This paper is generally well-written and easy to follow. The design principle and experimental evaluation are reasonable. The proposed ACM is compatible with existing merging methods. However, some weaknesses were pointed out by the reviewer, such as a marginal performance gain and unconvincing motivation.

The authors' rebuttal made the contribution of the paper clear to most reviewers. Compiling the reviewers' comments and discussion, I think that the strengths of this paper outweigh its weaknesses. Therefore, I recommend accepting this paper.

I would encourage the authors to include the explanations and experiments added in the rebuttal and discussion phase in the revised paper, and clarify the advantages and motivation of the presented method.